# PointMAC: Meta-Learned Adaptation for Robust Test-Time Point Cloud Completion

**Linlian Jiang**[1]    **Rui Ma**[2,4]    **Li Gu**[1]    **Ziqiang Wang**[1]
**Xinxin Zuo**[1*]    **Yang Wang**[1,3*]

[1]Concordia University    [2]Jilin University    [3]Mila - Quebec AI Institute
[4]Engineering Research Center of Knowledge-Driven Human-Machine Intelligence, MOE, China
{linlian.jiang, li.gu, ziqiang.wang}@mail.concordia.ca,
ruim@jlu.edu.cn, {xinxin.zuo, yang.wang}@concordia.ca

## Abstract

Point cloud completion is essential for robust 3D perception in safety-critical applications such as robotics and augmented reality. However, existing models perform static inference and rely heavily on inductive biases learned during training, limiting their ability to adapt to novel structural patterns and sensor-induced distortions at test time. To address this limitation, we propose PointMAC, a meta-learned framework for robust test-time adaptation in point cloud completion. It enables sample-specific refinement without requiring additional supervision. Our method optimizes the completion model under two self-supervised auxiliary objectives that simulate structural and sensor-level incompleteness. A meta-auxiliary learning strategy based on Model-Agnostic Meta-Learning (MAML) ensures that adaptation driven by auxiliary objectives is consistently aligned with the primary completion task. During inference, we adapt the shared encoder on-the-fly by optimizing auxiliary losses, with the decoder kept fixed. To further stabilize adaptation, we introduce Adaptive $\lambda$-Calibration, a meta-learned mechanism for balancing gradients between primary and auxiliary objectives. Extensive experiments on synthetic, simulated, and real-world datasets demonstrate that PointMAC achieves state-of-the-art results by refining each sample individually to produce high-quality completions. To the best of our knowledge, this is the first work to apply meta-auxiliary test-time adaptation to point cloud completion.

## 1 Introduction

Recent advances in 3D sensing have enabled safety-critical applications in autonomous driving [1], robotics [2], and AR [3], where reliable 3D perception is fundamental. Point clouds—the direct output of 3D sensors—are often incomplete due to occlusions, limited coverage, and sensor noise, severely impairing downstream tasks such as recognition, planning, and interaction. This highlights the urgent need for robust point cloud completion under diverse, unpredictable conditions.

Existing point cloud completion approaches largely adopt encoder–decoder architectures. Although recent works have introduced sophisticated decoders [4, 5, 6] with progressive refinement via localized expansion, performance remains constrained by the expressiveness of extracted features from the incomplete inputs. This has motivated the development of transformer-based models [7, 8, 9], which primarily enhance the encoder's ability to capture rich, contextual features by modeling global context through attention mechanism. However, despite their improved representation capacity achieved by scaling up the model, inference remains static at test time, regardless of the specific input point cloud. This rigidity constitutes a fundamental bottleneck: the inability to adjust internal representations per input, limiting the model's ability to leverage visible cues, particularly under novel occlusions

---

*Corresponding authors

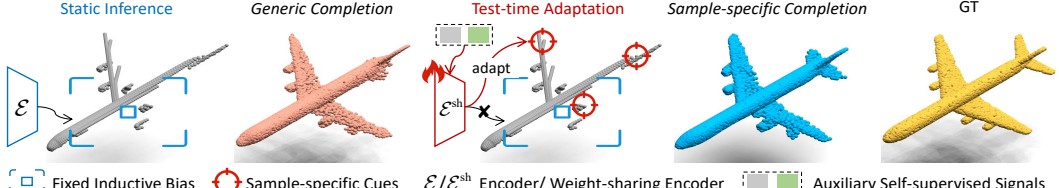

Figure 1: Existing point cloud completion models operate with fixed inductive biases **at inference**, often focusing on structurally stable regions (e.g., the fuselage). When such regions are missing, static inference hinders reasoning over other parts (e.g., the tail), resulting in *generic completions*. Instead, our model applies test-time adaptation using self-supervised signals, enabling dynamic attention to visible cues and producing *sample-specific completions* that better match the ground truth.

or sensor-induced distortions. As a result, the model tends to generate what we refer to as *generic completions*, which follow training priors and show limited sensitivity to input cues, rather than *sample-specific completions* that adapt to unique observations and restore geometry details (see Fig. 1). The generation of generic completions is further exacerbated by dataset limitations: synthetic data [4, 10] lacks structural diversity, while real-world scans [11] are limited in scale and coverage. These constraints reinforce inductive biases, causing the model to over-rely on structural priors and neglect sample-specific cues, ultimately degrading completion quality.

Motivated by these limitations, we shift from static inference to dynamic, sample-specific adaptation, enabling the model to refine predictions from each input's unique geometry and noise, leading to higher-quality completions. Test-time adaptation (TTA) offers a natural framework for this by enabling self-adaptation with unlabeled test data, and its effectiveness has been empirically validated [12, 13]. We thus treat each point cloud as a distinct domain, assuming that each input inherently reflects its source distribution. To this end, we introduce PointMAC, a TTA framework based on meta-auxiliary learning that performs per-sample refinement to improve completion accuracy.

First of all, PointMAC considers point cloud completion as the primary task and introduces an auxiliary branch, Bi-Aux Units, which performs self-supervised spatial-masking reconstruction and artifact denoising. Unlike conventional TTA methods [12, 13, 14] that optimize auxiliary losses in isolation, often resulting in misalignment with the primary task [15, 16], we adopt the Model-Agnostic Meta-Learning framework (MAML) [17] to regularize adaptation. Specifically, the auxiliary branches are optimized in the MAML inner loop to simulate the sample-specific adaptation, while the primary point cloud completion task supervises the outer loop to align the adaptation with the main objective. This meta-learning formulation encourages the model to adapt through optimizing auxiliary tasks in a way that directly benefits the primary task. At inference time, on-the-fly adaptation is performed by minimizing self-supervised losses defined using pseudo-labels generated by Bi-Aux Units. These losses are optimized via backpropagation, allowing the model to refine sample-specific outputs dynamically and overcome static attention patterns learned during training. To further stabilize adaptation and prevent task interference, we introduce Adaptive $\lambda$-Calibration, which harmonizes gradient contributions from the primary and auxiliary tasks during meta-training.

Our contributions can be summarized as follows:

- We propose PointMAC, a test-time adaptation method that addresses static encoder rigidity through sample-specific refinement, leveraging Bi-Aux Units to generate self-supervised signals under structural incompleteness and sensor noise for self-supervised adaptation.
- We introduce Adaptive $\lambda$-Calibration, a dynamic gradient balancing mechanism that mitigates negative transfer [15] during meta-training and improves test-time stability.
- To the best of our knowledge, this is the first application of meta-auxiliary learning and test-time adaptation in point cloud completion. PointMAC achieves state-of-the-art results on synthetic, simulated scanning, and real-world benchmarks, demonstrating strong generalization and adaptation capabilities across diverse point cloud domains.

## 2 Related Work

**Point Cloud Completion.** Recent approaches to point cloud completion have focused on architectural innovations [18, 5, 6, 9, 19]. However, most methods rely on static encoder features learned from

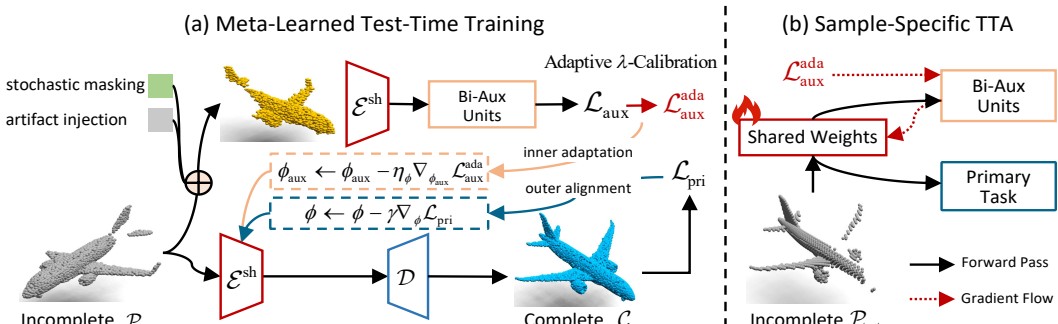

Figure 2: Overview of our test-time adaptation method. PointMAC formulates point cloud completion as the primary task and introduces Bi-Aux Units to provide self-supervised signals for test-time adaptation. The encoder $\mathcal{E}^{\mathrm{sh}}$ is shared between primary and auxiliary branches. In the meta-learned test-time training phase (a), sample-specific parameters are updated in the inner adaptation using auxiliary losses, while shared parameters are optimized in the outer alignment via the primary completion loss. In the sample-specific TTA phase (b), adaptation proceeds in three steps: (i) the meta-learned model produces initial completions; (ii) the shared encoder $\mathcal{E}^{\mathrm{sh}}$ is updated via self-supervised losses from Bi-Aux Units; (iii) sample-specific completions are generated, adapted to the unique structure and noise of each input.

biased synthetic datasets [4, 10], leading to inductive biases that generalize poorly to novel occlusions and sensor noise [11]. Without sample-specific adaptability at inference time, these models often yield generic completions that overlook input-specific cues and degrade reconstruction quality.

**Test-time Adaptation.** Test-time adaptation (TTA) addresses domain shift by adapting models online using unlabeled test data. Prior works have explored TTA across various domains, such as dynamic scene deblurring [20, 21], optical flow [22], and sequential modeling [23]. By leveraging self-supervised auxiliary signals from test inputs, TTA has demonstrated improved robustness and generalization [12, 24]. However, a key challenge lies in the misalignment between auxiliary and primary tasks, which can lead to unstable or suboptimal adaptation [15, 16]. To address this, we adopt the Model-Agnostic Meta-Learning (MAML) framework [17] to regularize adaptation.

**Meta Learning.** Meta-learning methods, such as Model-Agnostic Meta-Learning (MAML) [17], enable fast adaptation of pre-trained models to individual samples and have shown strong performance in few-shot learning [25, 26], as well as in auxiliary-task-guided multi-task training [27, 28]. Building on these advances [29, 30, 31], we incorporate meta-learning into our test-time adaptation framework to align self-supervised auxiliary objectives with the primary completion task, resulting in higher-fidelity and structure-aware shape completion.

## 3 Method

In this section, we introduce the proposed PointMAC method, including the network architecture and the test-time adaptation framework. As illustrated in Fig. 2, the overall architecture consists of a shared encoder, a primary decoder for shape reconstruction, and Bi-Aux Units that provide self-supervised signals by simulating structural occlusions and sensor-induced distortions, detailed in Sec. 3.1. To train the network, we adopt a meta-auxiliary learning strategy based on MAML to align the self-supervised auxiliary adaptation with the primary point cloud completion objective, enabling sample-specific adaptation at test time (detailed in Sec. 3.2).

### 3.1 Network Architecture

#### 3.1.1 Primary Branch

Given a partial and unordered point cloud $\mathcal{P} = \{\mathbf{p}_i\}_{i=1}^{M} \subset \mathbb{R}^3$, the goal of the primary task is to reconstruct a complete point cloud $\mathcal{C} = \{\mathbf{c}_j\}_{j=1}^{N} \subset \mathbb{R}^3$ with $N > M$. As illustrated in Fig. 2(a), we adopt a hierarchical encoder $\mathcal{E}^{\mathrm{sh}}$ inspired by [32, 33], to extract both local and global geometric features from $\mathcal{P}$, producing a compact shape code, $\mathbf{z} = \mathcal{E}^{\mathrm{sh}}(\mathcal{P}; \phi_{\mathrm{pri}}^{\mathrm{sh}})$. The decoder $\mathcal{D}$ takes $\mathbf{z}$ as

input and reconstructs the final point cloud $\mathcal{C} = \mathcal{D}(\mathbf{z}; \phi_{\text{pri}}^{\text{dec}})$ using a coarse-to-fine refinement strategy, following [19]. To supervise the primary task, we adopt the Chamfer Distance (CD) to measure the discrepancy between the predicted point cloud $\mathcal{C}$ and the ground truth $\mathcal{G}$. The CD between two point sets $\mathcal{X}, \mathcal{Y} \subset \mathbb{R}^3$ is defined as:

$$\mathcal{L}_{\text{CD}}(\mathcal{X}, \mathcal{Y}) = \frac{1}{|\mathcal{X}|} \sum_{x \in \mathcal{X}} \min_{y \in \mathcal{Y}} \|x - y\|_2^2 \; + \; \frac{1}{|\mathcal{Y}|} \sum_{y \in \mathcal{Y}} \min_{x \in \mathcal{X}} \|y - x\|_2^2, \tag{1}$$

where $|\mathcal{X}|$ and $|\mathcal{Y}|$ denote the number of points in each set. Based on Eq. (1), the primary loss is defined as $\mathcal{L}_{\text{pri}} = \mathcal{L}_{\text{CD}}(\mathcal{C}, \mathcal{G})$, where $\mathcal{C}$ and $\mathcal{G}$ are the predicted and ground-truth point clouds.

The encoder $\mathcal{E}^{\text{sh}}$ is shared between the primary branch for point cloud completion and the Bi-Aux Units (Sec. 3.1.2), and its parameters are denoted as $\phi_{\text{pri}}^{\text{sh}}$. We denote the full set of parameters for the primary task as $\phi_{\text{pri}}$. During test-time adaptation, we freeze the decoder $\mathcal{D}$ and update only the shared encoder $\mathcal{E}^{\text{sh}}$ to enable sample-specific feature refinement via auxiliary losses (Sec. 3.2).

### 3.1.2 Bi-Aux Units

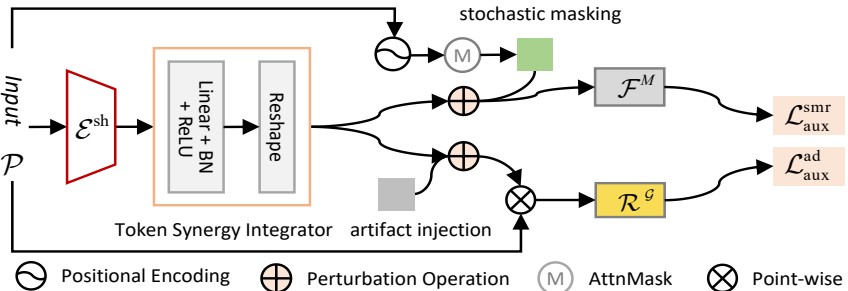

Figure 3: Overview of the proposed Bi-Aux Units, consisting of two self-supervised tasks—Stochastic Masked Reconstruction ($Aux^{\text{smr}}$) and Artifact Denoising ($Aux^{\text{ad}}$). Both branches share the encoder $\mathcal{E}^{\text{sh}}$ and Token Synergy Integrator $\mathcal{I}_{\text{TSI}}$ to ensure consistent feature conditioning, and output features ($\mathcal{F}^M, \mathcal{R}^{\mathcal{G}}$) that are projected to compute the auxiliary losses $\mathcal{L}_{\text{aux}}^{\text{smr}}$ and $\mathcal{L}_{\text{aux}}^{\text{ad}}$.

In addition to the primary branch, we introduce self-supervised Bi-Auxiliary (Bi-Aux) Units, which generate auxiliary signals to regularize the shared encoder during test-time training. Point cloud scans—whether synthetic or real—face two major challenges: **structural incompleteness** and **distortion robustness**. While synthetic data often exhibits regular missing patterns, real-world scans suffer from irregular occlusions and sensor noise, which exacerbate these issues. To address this, we design *Stochastic Masked Reconstruction* ($Aux^{\text{smr}}$) and *Artifact Denoising* ($Aux^{\text{ad}}$) to promote resilient encoder representations without ground-truth supervision.

**Stochastic Masked Reconstruction.** To mitigate structural bias in the shared encoder and improve robustness to diverse missing patterns, we design Stochastic Masked Reconstruction ($Aux^{\text{smr}}$, parameterized by $\phi_{\text{aux}}^{\text{smr}}$), which randomly masks spatial regions of the input cloud and trains the model to recover them.

As shown in Fig. 3, we apply Farthest-Point Sampling (FPS) to extract $N$ centroids $\mathcal{Q} = \{\mathbf{q}_k\}_{k=1}^N$ from the input cloud $\mathcal{P}$, and embed them into region tokens $\mathbf{z}_k = PE(\mathbf{q}_k) \in \mathbb{R}^D$ using a learnable positional encoder, while the shared encoder $\mathcal{E}^{\text{sh}}$ simultaneously generates global feature $\mathcal{F}$ from $\mathcal{P}$.

To reduce redundancy and enable parameter sharing across Bi-Aux Units, we introduce a *Token Synergy Integrator* ($\mathcal{I}_{\text{TSI}}$) with parameters $\phi_{\text{aux}}^{\text{sh}}$, which maps $\mathcal{F}$ into a group-token matrix $\mathcal{T}^{\mathcal{G}} \in \mathbb{R}^{N \times D}$ via an MLP stack (BN + ReLU) followed by reshaping. The transformation, shared across auxiliary signals via $\phi_{\text{aux}}^{\text{sh}}$, encourages consistent conditioning across tasks and eliminates redundant parameterization. This also supports effective test-time adaptation by enforcing shared representation priors across auxiliary tasks. The resulting group-token matrix $\mathcal{T}^{\mathcal{G}}$ is concatenated with region tokens $\{\mathbf{z}_k\}$, and the combined sequence is fed into dual-masked self-attention [34] to extract context-aware features:

$$\text{AttnMask}(Q, K, V; M) = \text{Softmax}\Big((QK^\top/\sqrt{D}) \; - \; (1 - M^d) \odot \infty\Big) V, \tag{2}$$

where $M^d$ is a binary mask applied to both rows and columns. The output features $\mathcal{F}^M$ are aggregated via max pooling along the token dimension to produce a compact latent vector, which is decoded via a lightweight [18] to reconstruct the complete point cloud $\widetilde{\mathcal{P}}$. The self-supervised loss is defined as: $\mathcal{L}_{\text{aux}}^{\text{smr}} = \mathcal{L}_{\text{CD}}(\widetilde{\mathcal{P}}, \mathcal{P})$. (see the Supplementary for further architectural details).

**Artifact Denoising.** Sensor-induced artifacts in real scans impede accurate shape recovery. Artifact Denoising ($Aux^{\text{ad}}$) mitigates this by corrupting input point clouds with realistic perturbations and training the model to restore clean geometry. This process encourages the encoder to learn distortion-resilient representations, enhancing robustness under real-world scanning conditions.

We introduce an auxiliary branch $Aux^{\text{ad}}$, parameterized by $\phi_{\text{aux}}^{\text{ad}}$, which performs artifact-aware denoising. This branch learns a mapping function $\Upsilon_{\varepsilon}^{\text{ad}} : \mathbb{R}^{M \times 3} \to \mathbb{R}^{\varepsilon M \times 3}$ (with $\varepsilon = 4$) to reconstruct a clean and dense point cloud $\widehat{\mathcal{P}} = \Upsilon_{\varepsilon}^{\text{ad}}(\overline{\mathcal{P}})$ from a noisy partial input $\overline{\mathcal{P}} = \mathcal{P} + \mathcal{N}(0, \sigma^2)$ (see Supplementary for details).

$Aux^{\text{ad}}$ builds on a shared architecture: it reuses the encoder $\mathcal{E}^{\text{sh}}$ from the primary branch to extract global features, and employs the shared Token Synergy Integrator $\mathcal{I}_{\text{TSI}}$ as described above in $Aux^{\text{smr}}$, to aggregate local context in a unified token space, yielding a refined sequence $\mathcal{R}^{\mathcal{G}}$. This design avoids duplicated learning efforts and facilitates effective cross-task knowledge transfer. Finally, we integrate the SpatialRefiner module from Dis-PU [35] to decode the features $\mathcal{R}^{\mathcal{G}}$ back to the original input point cloud. The output $\widehat{\mathcal{P}}$ is supervised by Chamfer Distance: $\mathcal{L}_{\text{aux}}^{\text{ad}} = \mathcal{L}_{CD}(\widehat{\mathcal{P}}, \mathcal{P})$.

## 3.2 Model Learning

Given the above network architecture, we will introduce our meta-auxiliary learning framework that allows sample-specific test-time adaptation in this section. Conventional TTA methods [27, 36] minimize auxiliary losses at test time, but misalignment with the primary task can lead to negative transfer [15]. PointMAC addresses this by leveraging first-order MAML [17] to align auxiliary updates with the primary objective.

**Meta-Learned TTA: *Training*.** As illustrated in Fig. 2(a), we enable effective test-time adaptation by simulating sample-specific encoder updates during training through interleaved meta-inner and outer loops. In the inner loop, the shared encoder and auxiliary branches are adapted using a single input point cloud randomly sampled from the training set. The outer loop then updates the full model to ensure that these auxiliary adaptations contribute to improving performance on the primary task. The full training procedure is detailed below.

**(i) Inner Auxiliary Adaptation**: We divide the model parameters into shared weights $\{\phi_{\text{pri}}^{\text{sh}}, \phi_{\text{aux}}^{\text{sh}}\}$ and sample-specific weights $\{\phi_{\text{pri}}, \phi_{\text{aux}}^{\text{smr}}, \phi_{\text{aux}}^{\text{ad}}\}$. For each auxiliary branch $a \in \{\text{smr}, \text{ad}\}$, we perform an inner-loop update at step $t$ by minimizing the auxiliary loss:

$$
\begin{aligned}
\phi_{\text{aux}}^{\text{smr}(t+1)} &\leftarrow \phi_{\text{aux}}^{\text{smr}(t)} - \alpha \cdot \nabla_{\phi_{\text{aux}}^{\text{smr}}} \mathcal{L}_{\text{aux}}^{\text{smr}} \left( \widetilde{\mathcal{P}}^{(t)}, \mathcal{P}^{(t)}; \phi_{\text{pri}}^{\text{sh}(t)}, \phi_{\text{aux}}^{\text{sh}(t)}, \phi_{\text{aux}}^{\text{smr}(t)} \right), \\
\phi_{\text{aux}}^{\text{ad}(t+1)} &\leftarrow \phi_{\text{aux}}^{\text{ad}(t)} - \beta \cdot \nabla_{\phi_{\text{aux}}^{\text{ad}}} \mathcal{L}_{\text{aux}}^{\text{ad}} \left( \widehat{\mathcal{P}}^{(t)}, \mathcal{P}^{(t)}; \phi_{\text{pri}}^{\text{sh}(t)}, \phi_{\text{aux}}^{\text{sh}(t)}, \phi_{\text{aux}}^{\text{ad}(t)} \right),
\end{aligned}
\tag{3}
$$

where $\widetilde{\mathcal{P}}$ and $\widehat{\mathcal{P}}$ are outputs of the two auxiliary branches, and $\alpha$, $\beta$ are their learning rates.

**(ii) Outer Primary Alignment**: Given the updated auxiliary task parameters, we align them with the primary objective by optimizing the primary task loss:

$$
\mathcal{L}_{\text{pri}} = \frac{1}{T} \sum_{i=1}^{T} \mathcal{L}_{\text{pri}}(\mathcal{C}^{(i)}, \mathcal{P}^{(i)}; \phi_{\text{pri}}),
\tag{4}
$$

where $T$ is the batch size, $\mathcal{L}_{\text{pri}}$ denotes the primary task loss function, and $\phi_{\text{pri}} = \{\phi_{\text{pri}}^{\text{sh}}, \phi_{\text{pri}}^{\text{dec}}\}$ represents the set of parameters for the primary task. The parameters are updated using gradient descent over the mini-batch, where $t$ denotes the current update step.

$$
\phi_{\text{pri}}^{(t+1)} \leftarrow \phi_{\text{pri}}^{(t)} - \gamma \cdot \nabla_{\phi_{\text{pri}}^{(t)}} \left( \frac{1}{T} \sum_{i=1}^{T} \mathcal{L}_{\text{pri}}(\mathcal{C}^{(i)}, \mathcal{P}^{(i)}; \phi_{\text{pri}}^{(t)}) \right),
\tag{5}
$$

where $\gamma$ is the learning rate for the primary task.

**Adaptive $\lambda$-Calibration.** Balancing multi-task losses is particularly brittle in test-time adaptation, where fixed weights can destabilize optimization or suppress the primary objective. While prior works [27, 29, 37] adopt static or manually tuned weights, such heuristics fail to generalize across samples or training stages. We propose Adaptive $\lambda$-Calibration, a meta-learned, gradient-based mechanism that dynamically adjusts the auxiliary weights $\lambda_{\mathrm{smr}}$ and $\lambda_{\mathrm{ad}}$ during training.

Specifically, the weights are softmax-normalized in logit space (Eq. (6)) and used to compute the total auxiliary loss in Eq. (7).

$$w_{\mathrm{smr}} = \left( \log(1 + \lambda_{\mathrm{smr}}^2) / \left[ \log(1 + \lambda_{\mathrm{smr}}^2) + \log(1 + \lambda_{\mathrm{ad}}^2) \right] \right), \quad w_{\mathrm{ad}} = 1 - w_{\mathrm{smr}}, \tag{6}$$

$$\mathcal{L}_{\mathrm{aux}}^{\mathrm{ada}} = w_{\mathrm{smr}} \cdot \mathcal{L}_{\mathrm{aux}}^{\mathrm{smr}} + w_{\mathrm{ad}} \cdot \mathcal{L}_{\mathrm{aux}}^{\mathrm{ad}}. \tag{7}$$

Both the auxiliary branch parameters $\phi_{\mathrm{aux}} = \{\phi_{\mathrm{aux}}^{\mathrm{smr}}, \phi_{\mathrm{aux}}^{\mathrm{ad}}\}$ and the weighting coefficients $\lambda \in \{\lambda_{\mathrm{smr}}, \lambda_{\mathrm{ad}}\}$ are jointly updated via gradient descent:

$$\phi_{\mathrm{aux}} \leftarrow \phi_{\mathrm{aux}} - \eta_\phi \nabla_{\phi_{\mathrm{aux}}} \mathcal{L}_{\mathrm{aux}}^{\mathrm{ada}}, \quad \lambda \leftarrow \lambda - \eta_\lambda \nabla_\lambda \mathcal{L}_{\mathrm{aux}}^{\mathrm{ada}}. \tag{8}$$

It jointly optimizes the model parameters of auxiliary branches and their relative weights in a task-aligned manner, allowing the model to automatically calibrate auxiliary signals based on their utility to the main objective.

**Sample-Specific TTA: *Inference*.** At inference, we perform a few self-supervised gradient steps on auxiliary losses for each test sample, as shown in Fig. 2(b). The auxiliary branches and their calibrated weights, learned during meta-training, remain fixed during adaptation. The shared encoder is refined by minimizing the combined auxiliary loss $\mathcal{L}_{\mathrm{aux}}^{\mathrm{ada}}$, as summarized in Alg. 1.

---

**Algorithm 1:** Sample-Specific Test-Time Adaptation

---

**Input:** Trained parameters $\phi = \{\phi_{\mathrm{pri}}^{\mathrm{sh}}, \phi_{\mathrm{pri}}, \phi_{\mathrm{aux}}^{\mathrm{sh}}, \phi_{\mathrm{aux}}^{\mathrm{smr}}, \phi_{\mathrm{aux}}^{\mathrm{ad}}\}$;
test input $\mathcal{P}_{\mathrm{test}}$; step size $\eta$; number of steps $K$
**Output:** Adapted encoder $\phi_{\mathrm{pri}}^{\mathrm{sh}}$
Apply stochastic masking and artifact injection using $\mathcal{M}$ and $\sigma$ via $Aux^{\mathrm{smr}}$ and $Aux^{\mathrm{ad}}$;
**for** $t = 0$ **to** $K - 1$ **do**                                     `/* Inner-loop adaptation */`
     $\widetilde{\mathcal{P}} \leftarrow Aux^{\mathrm{smr}}\_\texttt{Forward}(\mathcal{P}_{\mathrm{test}}, \mathcal{M}; \phi_{\mathrm{aux}}^{\mathrm{smr}})$;
     $\widehat{\mathcal{P}} \leftarrow Aux^{\mathrm{ad}}\_\texttt{Forward}(\mathcal{P}_{\mathrm{test}}, \sigma; \phi_{\mathrm{aux}}^{\mathrm{ad}})$;
     $\mathcal{L}_{\mathrm{aux}}^{\mathrm{ada}} \leftarrow \lambda_{\mathrm{smr}} \cdot \mathcal{L}_{\mathrm{aux}}^{\mathrm{smr}}(\widetilde{\mathcal{P}}, \mathcal{P}_{\mathrm{test}}) + \lambda_{\mathrm{ad}} \cdot \mathcal{L}_{\mathrm{aux}}^{\mathrm{ad}}(\widehat{\mathcal{P}}, \mathcal{P}_{\mathrm{test}})$;
     $\phi_{\mathrm{pri}}^{\mathrm{sh}(t+1)} \leftarrow \phi_{\mathrm{pri}}^{\mathrm{sh}(t)} - \eta \cdot \nabla_{\phi_{\mathrm{pri}}^{\mathrm{sh}}} \mathcal{L}_{\mathrm{aux}}^{\mathrm{ada}}$ ;      `/* Gradient descent on shared encoder */`
**return** $\phi_{\mathrm{pri}}^{\mathrm{sh}(K)}$

---

These label-free inner-loop updates perform sample-specific test-time adaptation, refining the encoder to better capture the visible structure and noise characteristics of each input. This improves feature quality, reduces completion error, and enhances generalization to novel inputs—all without supervision or retraining.

## 4 Experiments

In this section, we evaluate our method on three types of datasets: purely synthetic datasets (ShapeNet [10], PCN [4]), a high-fidelity simulated scanning dataset (MVP [38]), and a real-world scanned dataset (KITTI [11]). ShapeNet and PCN provide dense, uniformly sampled synthetic point clouds with paired ground truth for training and evaluation. MVP is a high-resolution, multi-view rendered benchmark that closely mimics real-world 3D scanning conditions, including occlusions and viewpoint variation. KITTI consists of real LiDAR scans with sparse, uneven points and no paired ground truth, and is used for evaluation only (see Supplementary for additional details related to Sec. 4).

### 4.1 Datasets and Evaluation Metric

**PCN.** The PCN dataset [4] includes 30,974 CAD models across eight categories. We evaluate reconstruction quality using the $\ell_1$-norm Chamfer Distance and follow prior works [4, 5, 6, 7, 8, 18, 19, 39] by using their released implementations and hyperparameters.

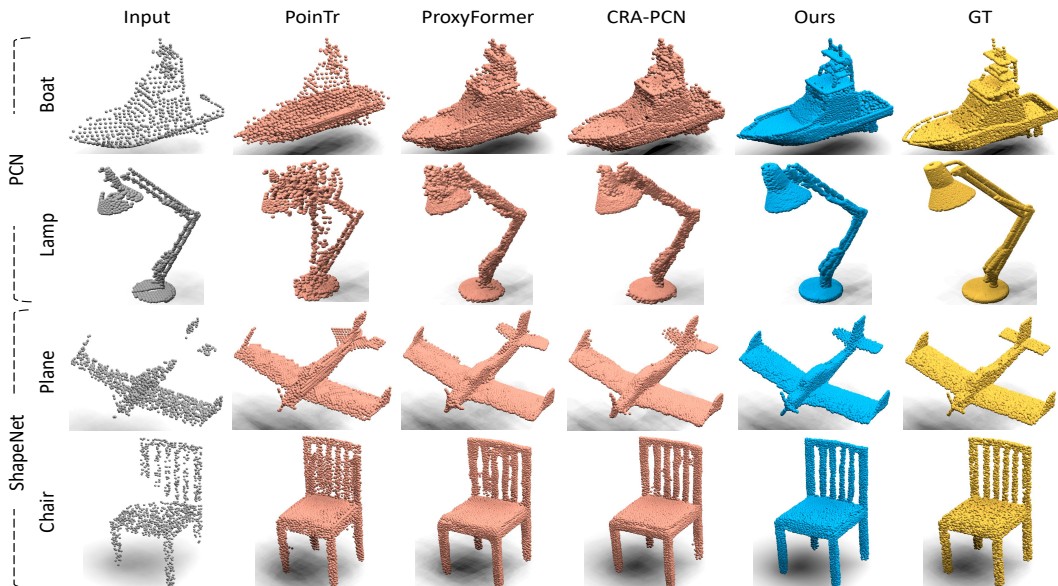

Figure 4: Visualization results on the PCN and ShapeNet datasets. Our method preserves fine-grained structures such as the complex geometry of boats and lamps, plane propellers and tails, and chair back slats, demonstrating strong completion quality and generalization across diverse categories.

Table 1: Quantitative comparison on the PCN dataset (per-point CD-$\ell_1$ × 1000). Both the output and ground truth point clouds consist of 16,384 points. (Lower CD is better)

| CD-$\ell_1$(×1000) | Plane | Cabinet | Car | Chair | Lamp | Couch | Table | Boat | CD-Avg |
|---|---|---|---|---|---|---|---|---|---|
| FoldingNet [18] | 9.49 | 15.80 | 12.61 | 15.55 | 16.41 | 15.97 | 13.65 | 14.99 | 14.31 |
| PCN [4] | 5.50 | 22.70 | 10.63 | 8.70 | 11.00 | 11.34 | 11.68 | 8.59 | 9.64 |
| SnowflakeNet [5] | 4.29 | 9.16 | 8.08 | 7.89 | 6.07 | 9.23 | 6.55 | 6.40 | 7.21 |
| PoinTr [7] | 4.75 | 10.47 | 8.68 | 9.39 | 7.75 | 10.93 | 7.78 | 7.29 | 8.38 |
| SeedFormer [6] | 3.85 | 9.05 | 8.06 | 7.06 | 5.21 | 8.85 | 6.05 | 5.85 | 6.74 |
| ProxyFormer [8] | 4.01 | 9.01 | 7.88 | 7.11 | 5.35 | 8.77 | 6.03 | 5.98 | 6.77 |
| EINet [39] | 3.96 | 8.81 | 7.74 | 6.93 | 5.03 | 8.80 | 6.15 | 5.57 | 6.63 |
| CRA-PCN [19] | 3.59 | 8.70 | 7.50 | 6.70 | 5.06 | 8.24 | 5.72 | 5.64 | 6.39 |
| **Ous** | **3.54** | **8.66** | **7.44** | **6.65** | **4.98** | **8.19** | **5.64** | **5.57** | **6.33** |

**ShapeNet-55/34.** Both datasets are derived from ShapeNet [10]. ShapeNet-55 provides 41,952 training and 10,518 testing shapes across 55 categories for category-agnostic evaluation. ShapeNet-34 offers 46,765 training shapes and 5,705 testing shapes from 34 categories, split into 3,400 seen-class and 2,305 unseen-class samples for category-specific generalization. Following standard protocol, we use Chamfer-$\ell_2$ distance and F-Score@1% [40] as evaluation metrics. Prior methods [4, 6, 7, 8, 18, 19, 39] are re-trained and evaluated under identical settings for fair comparison.

**MVP.** The MVP dataset [38] is a large-scale simulated scanning benchmark with over 100,000 partial-complete point cloud pairs across 16 categories. Partial shapes are rendered from 26 uniformly distributed views to simulate realistic occlusions. We use Chamfer-$\ell_2$ distance and F-Score@1% for evaluation, and compare with prior methods [4, 19, 38, 41, 42, 43, 44] under their official settings.

**KITTI.** We evaluate our method on the KITTI dataset [11], which consists of incomplete LiDAR-scanned car point clouds collected in real-world outdoor environments. Due to the lack of paired ground-truth shapes, we adopt Fidelity and Minimal Matching Distance (MMD) as evaluation metrics. Comparisons are conducted against prior methods [4, 6, 7, 8, 18, 39, 41, 45].

## 4.2 Evaluation on Main Datasets

**Results on PCN.** In Table 1, PointMAC achieves SOTA performance across all categories. Fig. 4 qualitatively compares our results with leading methods including PoinTr [7], ProxyFormer [8], and CRA-PCN [19]. Our method consistently generates more structurally coherent and high-fidelity

Table 2: Quantitative comparison on ShapeNet-55. We report CD-$\ell_2$ scores for the 10 major categories and the overall average across all 55 categories under three difficulty settings (CD-S, CD-M, CD-H for small, medium, hard), as well as the average F1 score. (Lower CD and higher F1 are better.)

| CD-$\ell_2$(×1000) | Table | Chair | Plane | Car | Sofa | Bird House | Bag | Remote | Key board | Rocket | CD-S | CD-M | CD-H | CD-Avg | F1 |
|---|---|---|---|---|---|---|---|---|---|---|---|---|---|---|---|
| FoldingNet [18] | 2.53 | 2.81 | 1.43 | 1.98 | 2.48 | 4.71 | 2.79 | 1.44 | 1.24 | 1.48 | 2.67 | 2.66 | 4.05 | 3.12 | 0.082 |
| PCN [4] | 2.13 | 2.29 | 1.02 | 1.85 | 2.06 | 4.50 | 2.86 | 1.33 | 0.89 | 1.32 | 1.94 | 1.96 | 4.08 | 2.66 | 0.133 |
| PoinTr [7] | 0.81 | 0.95 | 0.44 | 0.91 | 0.79 | 1.86 | 0.93 | 0.53 | 0.38 | 0.57 | 0.58 | 0.88 | 1.79 | 1.09 | 0.464 |
| SnowflakeNet [5] | 0.75 | 0.84 | 0.42 | 0.88 | 0.72 | 1.74 | 0.81 | 0.48 | 0.36 | 0.51 | 0.52 | 0.80 | 1.62 | 0.98 | 0.477 |
| SeedFormer [6] | 0.72 | 0.81 | 0.40 | 0.89 | 0.71 | - | - | - | - | - | 0.50 | 0.77 | 1.49 | 0.92 | 0.472 |
| ProxyFormer [8] | 0.70 | 0.83 | 0.34 | **0.78** | 0.69 | - | - | - | - | - | 0.49 | 0.75 | 1.55 | 0.93 | 0.483 |
| EINet [39] | 0.66 | 0.79 | 0.41 | 0.84 | 0.69 | 1.49 | 0.73 | 0.42 | 0.33 | 0.49 | 0.49 | 0.75 | 1.46 | 0.90 | 0.432 |
| CRA-PCN [19] | 0.66 | 0.74 | 0.37 | 0.85 | 0.66 | 1.36 | 0.73 | 0.43 | 0.35 | 0.50 | 0.48 | 0.71 | 1.37 | 0.85 | - |
| **Ous** | **0.65** | **0.72** | **0.34** | 0.80 | **0.64** | **1.34** | **0.72** | **0.40** | **0.31** | **0.47** | **0.47** | **0.69** | **1.34** | **0.83** | **0.490** |

Table 3: Quantitative comparison on Seen ShapeNet-34 test set and Unseen ShapeNet-21 test set. CD-$\ell_2$ for small, medium, and hard cases (CD-S, CD-M, CD-H) are reported (lower is better).

| CD-$\ell_2$(×1000) | 34 seen categories | | | | 21 unseen categories | | | |
|---|---|---|---|---|---|---|---|---|
| | CD-S | CD-M | CD-H | CD-Avg | CD-S | CD-M | CD-H | CD-Avg |
| FoldingNet [18] | 1.86 | 1.81 | 3.38 | 2.35 | 2.76 | 2.74 | 5.36 | 3.62 |
| PCN [4] | 1.87 | 1.81 | 2.97 | 2.22 | 3.17 | 3.08 | 5.29 | 3.85 |
| PoinTr [7] | 0.76 | 1.05 | 1.88 | 1.23 | 1.60 | 1.67 | 3.44 | 2.05 |
| SeedFormer [6] | 0.48 | 0.70 | 1.30 | 0.83 | 0.61 | 1.07 | 2.35 | 1.34 |
| ProxyFormer [8] | 0.44 | 0.67 | 1.33 | 0.81 | 0.60 | 1.13 | 2.54 | 1.42 |
| EINet [39] | 0.46 | 0.68 | 1.24 | 0.79 | 0.59 | 1.01 | 2.19 | 1.26 |
| CRA-PCN [19] | 0.45 | 0.65 | 1.18 | 0.76 | 0.55 | 0.97 | 2.19 | 1.24 |
| **Ours** | **0.44** | **0.64** | **1.14** | **0.75** | **0.53** | **0.96** | **2.16** | **1.22** |

completions, particularly in regions with fine-grained geometry. For instance, PointMAC reconstructs boats (first row) with smooth, continuous surfaces and clearly defined upper structures, while competing methods often produce broken or overly smoothed shapes lacking geometric sharpness. In the case of lamps (second row), our model faithfully recovers thin, articulated components such as the arm and head with precise alignment, whereas prior approaches frequently exhibit distortions, discontinuities, or missing parts in these complex areas. These results demonstrate the benefit of sample-specific refinement for geometric accuracy in challenging regions.

**Results on ShapeNet-55/34.** As shown in Table 2, PointMAC achieves SOTA performance on ShapeNet, demonstrating strong generalization across diverse object categories. Fig. 4 (last two rows) illustrates representative completions. For airplanes (third row), it accurately reconstructs fine structures such as propeller blades and tail fins, which prior methods often over-smooth or fragment. In the chair category (last row), our model recovers thin, densely arranged back slats with clear spacing and uniform thickness, while other approaches yield blurry slat structures and spurious points in unrelated areas (e.g., seat or legs). We also report results on the 34 seen categories of ShapeNet-34 in Table 3. On the 21 unseen categories, our method achieves the best overall performance. The consistent improvement across all difficulty levels and both seen and unseen subsets supports the core motivation of PointMAC: adapting to diverse structures and noise beyond training priors.

**Results on MVP.** Table 4 presents quantitative results on the MVP simulated scanning dataset. Point-MAC outperforms all baselines on both CD-$\ell_2$ and F-Score metrics. In particular, its advantage over CRA-PCN [19] suggests more accurate shape reconstruction and finer geometric detail preservation under realistic occlusions.

### 4.3 Cross-Dataset Evaluation

**Results on KITTI.** Since there are no paired groundtruth for KITTI, we train our model on ShapeNet-Cars [4] and evaluate it on KITTI. As shown in Table 5, PointMAC reduces the fidelity from 0.151 to 0.135 (a 10.6% relative reduction) and simultaneously decreases the MMD from 0.508 to 0.477, underscoring the effectiveness of sample-specific adaptation in handling real-world noise and recovering fine-grained geometry beyond training priors.

Table 4: Quantitative comparison on MVP dataset. We use CD-$\ell_2 \times 10^4$ and F1 Score for evaluation.

| | PCN [4] | TopNet [41] | MSN [42] | CDN [43] | ECG [44] | VRCNet [38] | CRA-PCN [19] | Ours |
|---|---|---|---|---|---|---|---|---|
| CD-$\ell_2$ ↓ | 9.77 | 10.11 | 7.90 | 7.25 | 6.64 | 5.96 | 5.33 | **5.24** |
| F1 ↑ | 0.320 | 0.308 | 0.432 | 0.434 | 0.476 | 0.499 | 0.529 | **0.537** |

Table 5: Quantitative comparison on the KITTI dataset. We use the Fidelity Distance and Minimal Matching Distance (MMD) for evaluation metrics. (Lower Fidelity and MMD are better)

| | PCN [4] | FoldingNet [18] | TopNet [41] | GRNet [45] | PoinTr [7] | SeedFormer [6] | ProxyFormer [8] | EINet [39] | Ours |
|---|---|---|---|---|---|---|---|---|---|
| Fidelity $\downarrow$ | 2.235 | 7.467 | 5.354 | 0.816 | **0.000** | 0.151 | **0.000** | 1.48 | 0.135 |
| MMD $\downarrow$ | 1.366 | 0.537 | 0.636 | 0.568 | 0.526 | 0.516 | 0.508 | 0.512 | **0.477** |

## 4.4 Ablation Studies

**Impact of Bi-Aux Units.** We conduct ablation studies on the PCN and ShapeNet-55 datasets. As shown in Table 6, (A) denotes the baseline, while (B) shows that integrating Bi-Aux Units significantly improves performance across both datasets. This suggests that the self-supervised auxiliary tasks provide reliable gradient signals, enabling the encoder to learn more robust and informative features that benefit the primary completion task and enhance overall completion quality.

Table 6: Ablation study on the PCN and ShapeNet-55 datasets. (A)–(C) examine the impact of the auxiliary design: (A) is the baseline; (B) adds Bi-Aux Units; (C) removes the Token Synergy Integrator. (D) is the full model without Adaptive $\lambda$-Calibration. (E) is our full framework with test-time adaptation.

| | Model | PCN ($\ell_1$) | ShapeNet ($\ell_2$) |
|---|---|---|---|
| (A) | Baseline | 6.62 | 0.90 |
| (B) | w/ Bi-Aux Units | 6.42 | 0.85 |
| (C) | w/o Token Synergy Integrator | 6.49 | 0.86 |
| (D) | w/o Adaptive $\lambda$-Calibration | 6.38 | 0.84 |
| (E) | Full Model | **6.33** | **0.83** |

We further observe that removing the shared weights in (C) Token Synergy Integrator disrupts the synergy between the primary and auxiliary tasks, thereby impeding effective information transfer.

**Impact of TTA Framework.** We evaluate the effectiveness of incorporating TTA with auxiliary signals. As shown in Table 6 (D), this strategy not only removes the need for costly manual tuning but also improves overall model performance. Furthermore, our full model (E) incorporates test-time adaptation with auxiliary signals. During inference, the model performs sample-specific adaptation for each input sample, enabling personalized predictions that better align with the unique characteristics of the input, as illustrated in Fig. 5. While jointly training with Bi-Aux Units (B) effectively improves the overall completion quality, it tends to produce **generic completion** (e.g., a straight lamp arm) that may overlook **sample-specific completion** (e.g., a circular ring). In contrast, our test-time adaptation strategy personalizes each input by refining the model's internal representation based on its unique geometry or noise pattern, moving beyond global statistical priors. This leads to more detailed and structurally faithful completions. As a result, the model is better able to preserve subtle, sample-specific structural cues that are often lost under static inference.

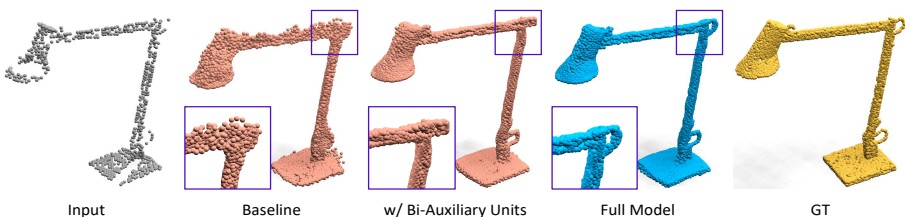

Figure 5: Visualization of the ablation study on different components of our framework.

## 5 Conclusion

We propose PointMAC, a sample-specific test-time training framework that mitigates the rigidity of static encoder attention via meta-auxiliary learning. Through dynamic inference refinement and stabilized adaptation, PointMAC effectively adapts to diverse and unseen scenarios. Experiments across synthetic, simulated, and real-world datasets demonstrate state-of-the-art performance, validating its robustness under occlusions and sensor noise.

**Limitations and Future Work.** Our experiments focus on single-object completion, but the framework can naturally extend to scene-level settings. Its label-free adaptation suits real-world scenarios like robotics and AR, where ground-truth supervision is costly. Future work includes scaling to complex environments and incorporating multi-modal guidance to enhance completion quality.

# Acknowledgements

This work was supported in part by NSERC, the National Natural Science Foundation of China (No. 62202199) and Science and Technology Development Plan of Jilin Province (No. 20230101071JC).

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

This supplementary document provides additional details to support the main paper.

In Section A, we elaborate on the implementation of our key components, including the stochastic masked reconstruction, artifact denoising, and the meta-auxiliary training strategy. Section B outlines the training configurations, hyperparameters, and implementation details. Section C presents additional qualitative results to further demonstrate the effectiveness and generalization ability of our method. Specifically, we include visual comparisons on both the MVP dataset [38] —a high-resolution, multi-view rendered simulation dataset—and the KITTI dataset [11], which contains real-world LiDAR scans without ground-truth supervision. These visualizations highlight the capability of our test-time adaptation framework to produce accurate, structurally coherent completions across diverse and challenging domains. Additionally, we provide additional completion examples on PCN [4] and ShapeNet [10], including zoom-in visualizations of local regions to demonstrate the ability of our method to recover fine-grained, sample-specific geometric details.

# A   Method Details

## A.1   Details of the Stochastic Masked Reconstruction

### A.1.1   Region Token and Global Feature Extraction

We first apply Farthest-Point Sampling (FPS) to the input point cloud $\mathcal{P} = \{\mathbf{p}_i\}_{i=1}^M$ to extract $N$ representative centroids $\mathcal{Q} = \{\mathbf{q}_k\}_{k=1}^N$. Each centroid $\mathbf{q}_k$ is then embedded into a region token using a learnable positional encoding module $PE(\cdot)$:

$$\mathbf{z}_k = PE(\mathbf{q}_k) \in \mathbb{R}^D, \tag{9}$$

In parallel, shared encoder $\mathcal{E}^{\mathrm{sh}}(\cdot)$ processes the full input point cloud $\mathcal{P}$ to extract per-point features. These are then aggregated using max pooling to produce a compact global feature vector:

$$\mathcal{F} = \mathrm{MaxPool}\big(\mathcal{E}^{\mathrm{sh}}(\mathcal{P})\big). \tag{10}$$

### A.1.2   Token Synergy Integrator

The Token Synergy Integrator ($\mathcal{I}_{\mathrm{TSI}}$) is designed to project the global feature vector into a token sequence that aligns with the spatially sampled region tokens. This transformation bridges global contextual information and local region-aware representations, enabling consistent conditioning across auxiliary tasks. The structure of the module is defined as:

$$\begin{aligned} \mathcal{I}_{\mathrm{TSI}}(\mathbf{x}) &= \mathrm{Reshape}\big(\mathrm{ReLU}\big(\mathrm{BN}\big(\mathrm{MLP}(\mathbf{x})\big)\big)\big), \\ \mathcal{T}^{\mathcal{G}} &= \mathcal{I}_{\mathrm{TSI}}\big(\mathrm{MaxPool}_{d=2}\big(\mathcal{E}^{\mathrm{sh}}(\mathcal{P})\big)\big). \end{aligned} \tag{11}$$

where $\mathrm{MaxPool}_{d=2}(\cdot)$ denotes 2D max pooling over the feature dimension. The MLP consists of two fully connected layers with intermediate dimensionality $2D$, and produces a flattened vector of length $N \cdot D$, which is reshaped to form the group-token matrix $\mathcal{T}^{\mathcal{G}} \in \mathbb{R}^{N \times D}$.

Since both auxiliary branches, $Aux^{\mathrm{smr}}$ and $Aux^{\mathrm{ad}}$, require task-specific token generators conditioned on the global feature, naively implementing separate modules would introduce redundant parameterization and potential inconsistency. To address this, we share the $\mathcal{I}_{\mathrm{TSI}}$ module across both auxiliary branches. This design encourages consistent representation learning and facilitates stable and efficient test-time adaptation. In our ablation studies, we observe that removing $\mathcal{I}_{\mathrm{TSI}}$ or breaking the parameter sharing leads to notable performance degradation, underscoring its importance in the overall framework.

### A.1.3   Masked Attention for Group–Region Token Fusion

The attention input is constructed by concatenating the group tokens $\mathcal{T}^{\mathcal{G}} \in \mathbb{R}^{N \times D}$ and region tokens $\{\mathbf{z}_k\} \in \mathbb{R}^{N \times D}$ along the token dimension, forming a sequence of $2N$ tokens with embedding

dimension $D$. All tokens are linearly projected using a shared projection matrix $W \in \mathbb{R}^{D \times D}$ to obtain the query, key, and value matrices:

$$Q = K = V = [\mathcal{T}^{\mathcal{G}}; \mathbf{Z}]\, W. \tag{12}$$

where $\mathbf{Z} = [\mathbf{z}_1; \ldots; \mathbf{z}_N] \in \mathbb{R}^{N \times D}$ denotes the stacked region tokens.

To control contextual interactions, we follow the dual-masking strategy from [34] and construct a binary attention mask $M^d \in \{0,1\}^{2N \times 2N}$ by independently sampling each element from a Bernoulli distribution. The mask is applied to both the row and column dimensions of the attention matrix and is shared across all attention heads.

After masked attention, the resulting feature sequence $\mathcal{F}^M \in \mathbb{R}^{2N \times D}$ is aggregated via max pooling along the token dimension to produce a compact latent representation $\mathbf{f}_{\text{latent}} \in \mathbb{R}^D$. This step not only compresses token-level representations but also reduces the computational load and parameter count for the subsequent decoder.

## A.2 Details of the Artifact Denoising

### A.2.1 Noise Injection

To better simulate sensor-induced imperfections observed in real-world 3D scans, we inject point-wise Gaussian noise into the input sparse point cloud $\mathcal{P} = \{\mathbf{p}_i\}_{i=1}^M$ prior to artifact-aware denoising. For each point $\mathbf{p}_i$, the noise standard deviation $\sigma_p$ is independently sampled from a uniform distribution:

$$\sigma_p \sim \mathcal{U}(0.001, 0.005), \tag{13}$$

representing $0.1\%$ to $0.5\%$ of the normalized coordinate scale. The perturbed point $\mathbf{p}_i'$ is then computed as:

$$\mathbf{p}_i' = \mathbf{p}_i + \boldsymbol{\epsilon}, \quad \boldsymbol{\epsilon} \sim \mathcal{N}(0, \sigma_p^2 \mathbf{I}). \tag{14}$$

where $\mathbf{I} \in \mathbb{R}^{3 \times 3}$ is the identity matrix, implying isotropic Gaussian noise added independently to each coordinate axis. To avoid extreme perturbations, each dimension of $\boldsymbol{\epsilon}$ is clipped to the range $[-0.02, 0.02]$. The resulting noisy input $\overline{\mathcal{P}} = \{\mathbf{p}_i'\}_{i=1}^M$ is processed by the auxiliary denoising branch $Aux^{\text{ad}}$, which applies the mapping $\Upsilon_\varepsilon^{\text{ad}} : \mathbb{R}^{M \times 3} \to \mathbb{R}^{\varepsilon M \times 3}$ to reconstruct a clean and dense point cloud $\widehat{\mathcal{P}} = \Upsilon_\varepsilon^{\text{ad}}(\overline{\mathcal{P}})$.

This procedure captures the heterogeneous and spatially varying noise distributions commonly encountered in practical 3D acquisition scenarios, and supports robust refinement during TTA.

## A.3 Details of the Model Learning

### A.3.1 Learning a Meta-Initialization via Joint Optimization

Joint training assumes that auxiliary tasks consistently benefit the primary objective. However, under distribution shift, static optimization can lead to gradient conflict and negative transfer, where updates from auxiliary tasks misalign with the primary goal. To address this, we adopt MAML [17] to meta-optimize the shared parameters, ensuring that adaptations guided by auxiliary losses consistently improve primary performance. Specifically, we first jointly optimize the primary and auxiliary objectives on source-domain data to obtain a generalizable initialization. Let $\mathcal{L}_{\text{pri}}$, $\mathcal{L}_{\text{aux}}^{\text{smr}}$, and $\mathcal{L}_{\text{aux}}^{\text{ad}}$ denote the primary and auxiliary task losses, respectively. During this joint training stage, we simultaneously minimize a weighted sum of the two objectives:

$$\mathcal{L} = \mathcal{L}_{\text{pri}}(\mathcal{C}, \mathcal{P}; \phi_{\text{pri}}^{\text{sh}}, \phi_{\text{pri}}) + \mu \Big[ \mathcal{L}_{\text{aux}}^{\text{smr}}(\widetilde{\mathcal{P}}, \mathcal{P}; \phi_{\text{pri}}^{\text{sh}}, \phi_{\text{aux}}^{\text{sh}}, \phi_{\text{aux}}^{\text{smr}}) + \mathcal{L}_{\text{aux}}^{\text{ad}}(\widehat{\mathcal{P}}, \mathcal{P}; \phi_{\text{pri}}^{\text{sh}}, \phi_{\text{aux}}^{\text{sh}}, \phi_{\text{aux}}^{\text{ad}}) \Big]. \tag{15}$$

where $\mu \in (0,1]$ balances the supervised primary loss and the two self-supervised auxiliary losses. This process updates all shared parameters to encourage learning representations that are both effective for the primary task and guided by complementary auxiliary signals.

The resulting parameters $\{\phi_{\text{pri}}^{\text{sh}}, \phi_{\text{pri}}\}$ serve as a warm-start for the subsequent meta-optimization phase, providing a stable initialization that incorporates task-relevant structures learned from both objectives.

### A.3.2 Meta-Learned TTA: Training Details

To align the auxiliary tasks with the primary objective under distribution shift, we adopt a meta-auxiliary training procedure that dynamically adjusts the contribution of each auxiliary loss during training. The goal is to ensure that updates guided by auxiliary signals consistently improve the primary performance, thereby yielding a better initialization for test-time adaptation.

Let $\phi = \{\phi_{\text{pri}}^{\text{sh}}, \phi_{\text{pri}}, \phi_{\text{aux}}^{\text{sh}}, \phi_{\text{aux}}^{\text{smr}}, \phi_{\text{aux}}^{\text{ad}}\}$ denote the full set of parameters, and $\phi_{\text{aux}}^{\text{smr}}, \phi_{\text{aux}}^{\text{ad}}$ denote the task-specific auxiliary heads for stochastic masked reconstruction and artifact denoising. We initialize the adaptive auxiliary weights $(\lambda_{\text{smr}}, \lambda_{\text{ad}})$ in logit space and update them jointly with the model parameters in each iteration. Given a mini-batch $\{\mathcal{P}^{(t)}, \mathcal{C}^{(t)}\}_{t=1}^T$, we first compute the auxiliary losses $\mathcal{L}_{\text{aux}}^{\text{smr}}$ and $\mathcal{L}_{\text{aux}}^{\text{ad}}$. These losses are then combined into a single adaptive auxiliary objective using normalized task-specific weights, obtained via a softmax-like function. Both the auxiliary network parameters and the weighting coefficients are updated through gradient-based optimization. Finally, the primary loss is used to update the full model, ensuring that the shared encoder is guided by auxiliary signals that consistently support the primary objective. This process produces an initialization better suited for downstream test-time adaptation. The complete process is illustrated in Alg. 2.

---

**Algorithm 2:** Meta-Auxiliary Training

**Input:** Parameters $\phi = \{\phi_{\text{pri}}^{\text{sh}}, \phi_{\text{pri}}, \phi_{\text{aux}}^{\text{sh}}, \phi_{\text{aux}}^{\text{smr}}, \phi_{\text{aux}}^{\text{ad}}\}$; learning rates $\eta_\phi, \eta_\lambda, \gamma$
**Output:** meta-trained weights $\phi$, calibrated $(\lambda_{\text{smr}}, \lambda_{\text{ad}})$

Initialise $\lambda_{\text{smr}}, \lambda_{\text{ad}} \leftarrow 0$ (logit space);
**while** *not converged* **do**

 sample mini-batch $\{\mathcal{P}^{(t)}, \mathcal{C}^{(t)}\}_{t=1}^T$ ;       $\triangleright$ *auxiliary forward*
 Evaluate the auxiliary losses $\mathcal{L}_{\text{aux}}^{\text{smr}}, \mathcal{L}_{\text{aux}}^{\text{ad}}$ ;      $\triangleright$ *$\lambda$ normalisation*
 $\tilde{\alpha} \leftarrow \log(1 + \lambda_{\text{smr}}^2), \tilde{\beta} \leftarrow \log(1 + \lambda_{\text{ad}}^2)$;
 $w_{\text{smr}} \leftarrow \exp(\tilde{\alpha})/(\exp(\tilde{\alpha}) + \exp(\tilde{\beta})), w_{\text{ad}} \leftarrow 1 - w_{\text{smr}}$;
 $\mathcal{L}_{\text{aux}}^{\text{ada}} \leftarrow w_{\text{smr}}\mathcal{L}_{\text{aux}}^{\text{smr}} + w_{\text{ad}}\mathcal{L}_{\text{aux}}^{\text{ad}}$ ;     $\triangleright$ *update aux branch*
 $\phi_{\text{aux}} \leftarrow \phi_{\text{aux}} - \eta_\phi \nabla_{\phi_{\text{aux}}} \mathcal{L}_{\text{aux}}^{\text{ada}}$ ;      $\triangleright$ *update weights*
 $(\lambda_{\text{smr}}, \lambda_{\text{ad}}) \leftarrow (\lambda_{\text{smr}}, \lambda_{\text{ad}}) - \eta_\lambda \nabla_\lambda \mathcal{L}_{\text{aux}}^{\text{ada}}$.
 $\phi \leftarrow \phi - \gamma \nabla_\phi \mathcal{L}_{\text{pri}}$ ;         $\triangleright$ *outer update*

---

Alg. 2 dynamically calibrates the auxiliary loss weights $(\lambda_{\text{smr}}, \lambda_{\text{ad}})$ via per-iteration normalization and gradient-based updates. To prevent task misalignment and ensure that auxiliary supervision consistently benefits the primary task, we embed this process into a meta-learning framework based on MAML. In this formulation, the primary point cloud completion task supervises the optimization of auxiliary branches through outer-loop gradients, enabling the model to leverage auxiliary signals for more effective, sample-specific adaptation.

### A.4 Number of Gradient Updates

The number of gradient updates in the inner loop is a critical hyperparameter in our meta-auxiliary optimization framework. Existing test-time adaptation methods [30] typically adopt a fixed and limited number of updates without systematically analyzing its impact on adaptation quality. However, insufficient updates may result in under-adaptation to the target distribution, whereas excessive updates can lead to overfitting on auxiliary tasks.

To investigate this trade-off, we evaluate our method with different update steps $K \in \{1, 3, 5\}$, and report the results in Table 7. All experiments are conducted on the PCN [4] and ShapeNet [10] datasets, with the number of gradient steps kept consistent between training and testing.

As shown in Table 7, our method outperforms state-of-the-art approaches [8, 6, 39, 19] even with only three gradient updates ($K = 3$), and further improves with five updates ($K = 5$). These

Table 7: Ablation on update steps: performance under different numbers of gradient updates ($= 1, 3, 5$) on PCN and ShapeNet. Lower is better.

| | Model | PCN ($\ell_1 \downarrow$) | ShapeNet ($\ell_2 \downarrow$) |
|---|---|---|---|
| (A) | w/ Bi-Aux Units | 6.42 | 0.85 |
| (B) | w/ Bi-Aux Units + TTA ($K = 1$) | 6.40 | 0.84 |
| (C) | w/ Bi-Aux Units + TTA ($K = 3$) | 6.33 | 0.83 |
| (D) | w/ Bi-Aux Units + TTA ($K = 5$) | **6.28** | **0.81** |

results underscore the effectiveness of our test-time adaptation strategy in enabling high-quality, sample-specific completions through minimal per-instance optimization.

To balance accuracy and computational efficiency, we set $K = 3$ in all subsequent experiments. Investigation of larger update steps (e.g., $K = 7$) is left for future work.

## B    Implementation Details

We train the model for 250 epochs on the PCN [4] and ShapeNet [10] datasets, and for 200 epochs on MVP [38]. The batch size is set to 40 for PCN, 32 for ShapeNet, and 44 for MVP. During the joint training phase, we apply equal learning rates for the primary and auxiliary branches, with $\alpha = \beta = 2.5 \times 10^{-5}$.

In the meta-training and meta-testing stages, we perform 3 inner-loop gradient update steps to adapt the shared encoder using the auxiliary losses $\mathcal{L}_{\text{aux}}^{\text{smr}}$ and $\mathcal{L}_{\text{aux}}^{\text{ad}}$. Optimization is carried out using Stochastic Gradient Descent (SGD) without momentum or weight decay. All experiments are conducted on two NVIDIA V100 GPUs.

## C    Visualization

To further demonstrate the strong generalization ability of our test-time adaptation framework, we present additional qualitative results on MVP [38] (Fig.6) and KITTI[11] (Fig.7). MVP is a high-resolution, multi-view rendered benchmark that closely mimics real-world 3D scanning conditions, while KITTI consists of real-world LiDAR scans without ground-truth supervision.

We compare our method with several state-of-the-art completion approaches, including PoinTr[7], ProxyFormer [8], and CRA-PCN [19]. Across both datasets, our method consistently produces more complete and detail-preserving reconstructions. Notably, unlike PoinTr and ProxyFormer, which often generate over-smoothed outputs and fail to adapt to input-specific cues, our method preserves fine-grained structures and sharp object boundaries. Compared to CRA-PCN, our approach yields cleaner contours and fewer noisy artifacts, especially in complex or partially occluded regions. These results highlight the strength of our dynamic, sample-specific adaptation strategy and its robustness across both synthetic and real-world domains without relying on ground-truth supervision.

In addition, we provide more completion results on samples from PCN [4] and ShapeNet [10] (Fig. 8), including zoomed-in visualizations of local regions. These results further demonstrate the capability of our method to generate sample-specific completions by adapting to the unique structure of each input. Notably, our approach restores fine-grained details—such as thin bars, wings, and structural frames—across a wide range of categories, highlighting the benefit of dynamic refinement over static inference.

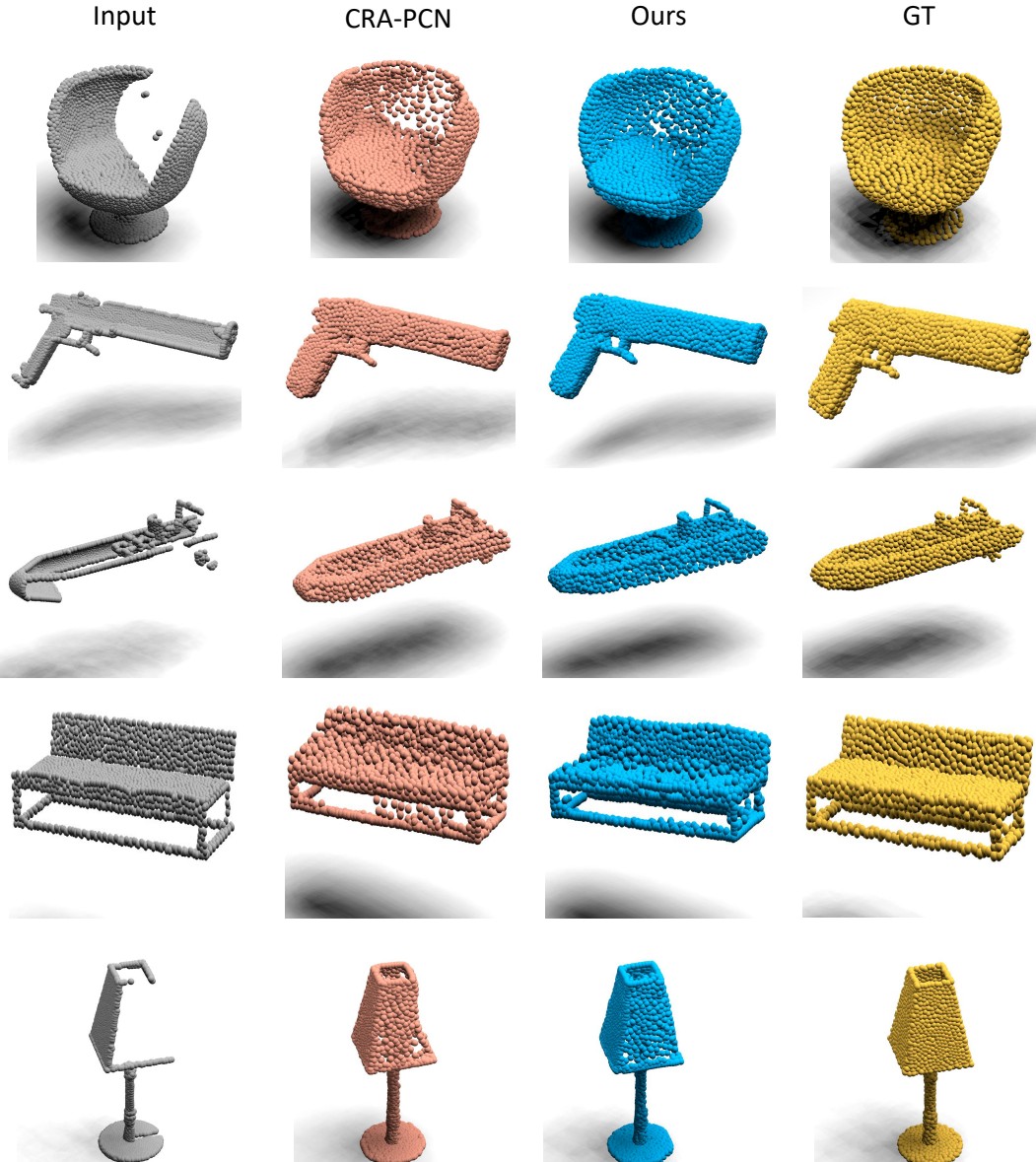

Figure 6: Qualitative comparison of point cloud completion results on the MVP dataset [38]. From left to right: incomplete input, results from CRA-PCN [19], our method, and ground truth. Compared to CRA-PCN, our completions present clearer structures and finer details with notably less contour noise. Specifically, our method better reconstructs critical regions across different categories: (row 1) the curved backrest of the chair, (row 2) the trigger area of the gun, (row 3) the complex hull structure of the boat, (row 4) the fine-grained details of the bench, and (row 5) the geometry of the lampshade. These results demonstrate the effectiveness of our test-time adaptation approach, which dynamically extracts sample-specific information to produce structurally accurate and detail-preserving completions.

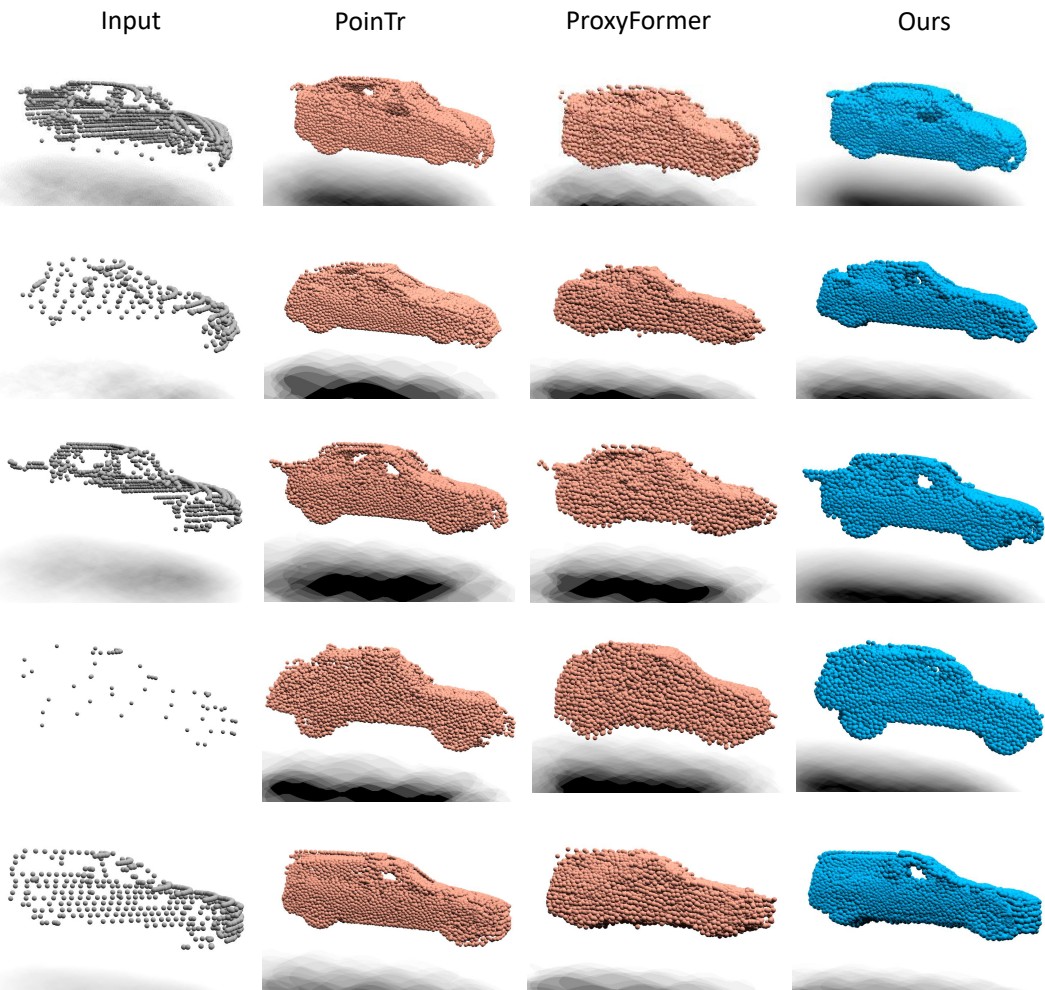

Figure 7: Qualitative comparison of point cloud completion results on the KITTI dataset [11]. From left to right: incomplete input, completion by PoinTr [7], ProxyFormer [8], our method, and ground truth. Among existing methods, ProxyFormer achieves the strongest overall performance; however, its outputs remain generic and often overly smoothed, obscuring fine-grained structures. PoinTr similarly struggles to preserve input-specific geometric details. In contrast, our method performs dynamic, per-sample refinement via test-time adaptation, resulting in more accurate and detailed completions. Notably, our results retain sharp object boundaries and recover distinctive features such as car windows and wheels, demonstrating superior geometric fidelity. Importantly, despite the absence of ground-truth during inference on real-world KITTI data, our approach leverages self-supervised signals to adapt effectively to each input, producing semantically meaningful and structurally consistent completions.

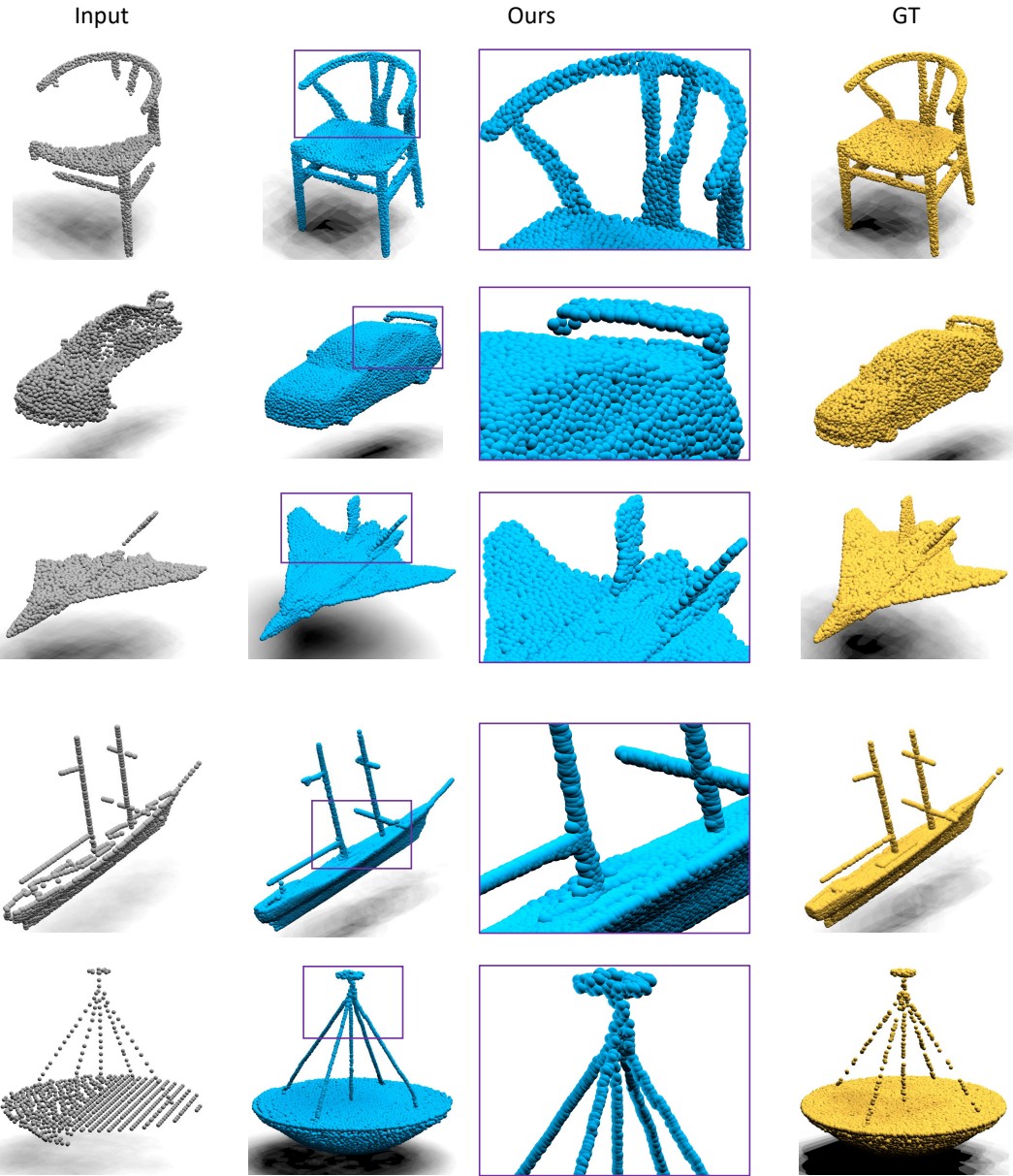

Figure 8: Qualitative results of point cloud completion on samples from the PCN [4] and ShapeNet [10] datasets. From left to right: incomplete input, our completion result, a zoom-in of the region highlighted by the purple bounding box, and ground truth. The examples cover diverse categories such as chairs, cars, airplanes, ships, and lamps. Unlike existing approaches that tend to produce generic completions, our method dynamically adapts to each input and faithfully reconstructs both global object structures and fine-grained details—such as the curved backrest bars of chairs, car rear wings, airplane tails, ship masts, and lamp frames. The zoomed-in views clearly demonstrate the effectiveness of our test-time adaptation strategy in preserving geometric fidelity and restoring delicate, sample-specific structural elements.

