# OpenReview forum: "PointMAC: Meta-Learned Adaptation for Robust Test-Time Point Cloud Completion"
_NeurIPS.cc/2025/Conference — NeurIPS 2025 poster_

### Official Review · Reviewer_FjXF · 2025-06-29

**Clarity:** 3
**Significance:** 2
**Originality:** 3
**Rating:** 4
**Confidence:** 3

**Summary:**

In this paper, the authors proposed Bi-Aux Units, which include two self-supervised tasks, Stochastic Masked Reconstruction and Artifact Denoising, and adopted a MAML-based meta-auxiliary learning strategy to align the auxiliary tasks with the primary point cloud completion objective, so that only the shared encoder can be adjusted during the test time to perform sample-specific adaptation for each sample. Experiments on large-scale synthetic, simulated, and real datasets verify the leading performance of this method.

**Questions:**

Q1. The coverage of the robustness evaluation of PointMAC is not clear. This paper proposes Stochastic Masked Reconstruction and Artifact Denoising to enhance the robustness of the encoder for point cloud completion tasks. The experiment lacks analysis of interference such as extreme occlusion and different noise intensities.

Q2. The explanation of the generalization ability of PointMAC is insufficient. Please further explain the effectiveness of the applied TTA mechanism in cross-dataset/cross-category scenarios. The article only uses ShapeNet-Cars for training and tests on KITTI, which is not enough to show that the model can avoid overfitting and maintain stable performance. It is recommended to further supplement cross-dataset experiments.

Q3. Some design details are not explained. Please explain the specific motivation and rationality of choosing Farthest-Point Sampling (FPS) instead of other sampling methods in Stochastic Masked Reconstruction, and using Gaussian noise instead of other noise to simulate sensor artifacts in Artifact Denoising. Please explain in detail how the learnable positional encoder in Stochastic Masked Reconstruction is designed and trained. At the same time, the impact of adaptive steps, learning rate and other hyperparameters on stability and efficiency during testing is not discussed in depth, making it difficult to fully understand the design motivation and decision.

Minor issues: In section 4.3 Cross-Dataset Evaluation, the text says that CD-l2 and F1 are used, but Table 5 lists Fidelity and MMD. Please confirm whether the descriptions of the two indicators are consistent.

**Ethical Concerns:**

["NO or VERY MINOR ethics concerns only"]

**Final Justification:**

The authors added an additional experiment on cross dataset experiment, which somehow validates the robustness of the proposed method.

**Limitations:**

The work listed the limitation, but only gave the future possible direction that can be easily extended. It is less informative.

**Paper Formatting Concerns:**

None.

**Quality:**

3

**Strengths And Weaknesses:**

S1. It proposes a practical solution to the problem that point clouds are incomplete due to occlusion and noise in real scenes, which is significatn.

S2. The proposed MAML-based strategy is intuitive.

W1. Insufficient evaluation of PointMAC robustness. The current experiment only examines limited noise intensity and occlusion ratio, lacks a systematic evaluation of heavy occlusion and abnormal noise distribution scenes, and it is difficult to prove the stability of the model under extreme real-world conditions.

W2. Insufficient evaluation of PointMAC generalization. In addition to testing KITTI after ShapeNet-Cars training, the paper does not show a systematic comparison across categories and dataset domains, so it cannot fully demonstrate that the method can fully handle real-world noise.

W3. Insufficient interpretability of some modules in the PointMAC design architecture and lack of explanation for hyperparameters. For example, the learnable positional encoder in Bi-Aux Units is not explained, and key hyperparameters such as the number of adaptive steps and learning rate during testing are not given.

---

> ### Author Rebuttal · Authors · 2025-07-30
>
> We thank Reviewer FjXF for the valuable comments. Specific concerns are also addressed below.
>
> # Q1: Robustness Evaluation Coverage
> ## Q1-1: Extreme Occlusion Evaluation
> We thank the reviewer for highlighting this important aspect. We explicitly evaluate the robustness of our approach on both synthetic and real-world datasets under severe structural incompleteness and sensor degradation. **Visualizations of extreme patterns are provided in Supplementary Sec. C (Line 911).**
> * On ShapeNet **(synthetic)**, we tested with up to 75\% of input points missing across 55 categories, as reported in Tables 2–3 of the main paper (CD-H metric).
> * On KITTI **(real-world)**, the input is even sparser (averaging 440 points, or 5.4\% of the completion target, with some samples below 10 points), where our model still achieves strong performance.
> ## Q1-2: Noise-Level Diversity
> We inject varying noise intensities to improve robustness **(see Supplementary Sec. A.2, Lines 816-829)**. Furthermore, KITTI scans naturally contain a wide range of noise intensities, density, and missing regions caused by sensor and environmental factors, making it an ideal benchmark for evaluating robustness.
> ## Q1-3: Benchmark Coverage
> To the best of our knowledge, our approach is among the most thoroughly evaluated, achieving SOTA on **four major benchmarks** (PCN, ShapeNet, MVP, KITTI), while most **prior works report on only two or three**. We welcome suggestions and are willing to conduct additional experiments as suggested.
> # Q2: Cross-dataset Experiment
> In addition to KITTI, we evaluated additional cross-dataset experiments with the PCN-pretrained model on Completion3D dataset[1]:
>
> **Rebuttal Table A.** Cross-dataset Results on Completion3D.
>
> | CD (ℓ₂)      | Plane | Cabinet | Car  | Chair | Lamp | Couch | Table | Boat | Avg   |
> |--------------|-------|---------|------|-------|------|-------|-------|------|-------|
> | ProxyFormer  | 4.15  | 13.53   | 7.81 | 8.44  | 7.01 | 11.73 | 7.28  | 5.05 | 8.12  |
> | CRA-PCN      | 4.03  | 13.36   | 7.60 | 7.92  | 6.33 | 11.21 | 6.82  | 4.29 | 7.70  |
> | Ours         | 3.77  | 12.49   | 7.17 | 7.42  | 5.86 | 10.54 | 6.32  | 3.85 | 7.18  |
>
> # Q3: Explanation of Design Details
> ## Q3-1: Choice of Farthest-Point Sampling (FPS)
> We adopt FPS as the **default masking strategy** in this field, following a well-established standard [2-4], which has shown FPS produces more uniform and globally distributed missing regions than random or grid-based sampling. This characteristic is especially beneficial for our point cloud completion task, enabling the model to better infer unseen parts when facing missing regions.
> ## Q3-2: Choice of Gaussian Noise
> We choose Gaussian noise because it is the **standard choice for simulating sensor artifacts** in point cloud learning [5-6]. Measurement errors in real-world 3D sensors are well-approximated by a Gaussian distribution (per the central limit theorem).
> ## Q3-3: Positional Encoder
> The learnable positional encoder is implemented as an **MLP** that maps each patch center’s spatial coordinate into a high-dimensional embedding, a technique **widely used in point cloud representation learning** [3] [7].
>
> The positional encoder, together with all other components of the Stochastic Masked Reconstruction module, is trained jointly in an end-to-end manner by minimizing the masked reconstruction loss **(see Supplementary Sec. A1, Lines 780–815)**.
> ## Q3-4: Key Hyperparameters
> Please see **Supplementary Sections A3–B (Lines 830–890)** for details of key hyperparameters and their impact on stability and efficiency. We will move this key discussion into the main text in the final version.
>
> **[Dear Reviewer FjXF]** We sincerely appreciate the reviewer’s attention to detail and high standards. We value your feedback and look forward to addressing any further questions or concerns.
>
> [1] Topnet: Structural point cloud decoder.
>
> [2] Pointnet++: Deep hierarchical feature learning on point sets in a metric space.
>
> [3] Masked Autoencoders for Point Cloud Self-supervised Learning.
>
> [4] Point transformer.
>
> [5] PointCleanNet: Learning to Denoise and Remove Outliers from Dense Point Clouds.
>
> [6] Point cloud denoising review: from classical to deep learning-based approaches.
>
> [7] Point-BERT: Pre-training 3D Point Cloud Transformers with Masked Point Modeling.

---

> > ### Comment · Reviewer_FjXF · 2025-08-04
> >
> > The reviewer thanks the rebuttal response of the authors. The cross dataset experiment does address part of my concerns. I will raise my score correspondingly.

---

> ### Author Response · Authors · 2025-08-07
>
> **Dear Reviewer FjXF**, thank you very much for your constructive suggestions and for raising the rating. We truly value your recognition and support. We are grateful that your concerns have been addressed and would be pleased to discuss any further concerns you may have.

---

### Official Review · Reviewer_PV7J · 2025-06-30

**Clarity:** 3
**Significance:** 3
**Originality:** 3
**Rating:** 4
**Confidence:** 3

**Summary:**

This paper proposes PointMAC, a novel meta-learning framework for robust test-time adaptation in point cloud completion, aiming to address the limitations of existing models that perform static inference and rely heavily on training-learned inductive biases. Extensive experiments on several benchmarks show its effectiveness.

**Questions:**

Refer to the Weakness.

**Ethical Concerns:**

["NO or VERY MINOR ethics concerns only"]

**Final Justification:**

The authors addressed my concerns and questions.  I will maintain my rating

**Limitations:**

The author analyzes the limitations and societal impacts adequately.

**Quality:**

3

**Strengths And Weaknesses:**

Goodness:

1.	The work introduces a novel application of meta-auxiliary learning and test-time adaptation (TTA) to point cloud completion, to address the static inference limitations of prior models.

2.	This paper accounts for the the static inference limitations of prior models, particularly under novel occlusions or sensor-induced distortions. PointMAC directly tackles this problem with sample-specific adaptation, enabling the model to refine predictions from each unique geometry and noise of input, leading to high-quality completions.

3.	Extensive experiments demonstrate its effectiveness.

4.	This paper is written and organized well.

Weakness:

1.	Since the proposed Bi-Aux Units module has the ability to learn the specific part of the input, it will be interesting to add more visualization results of the simulated results.

2.	Test-time adaptation (TTA) inherently involves performing several gradient update steps for each sample during inference. This inevitably increases the per-sample inference time. It will be interesting to report on the computational cost impact of this component.

3.	The proposed Bi-Aux Units consist of two specific self-supervised tasks: stochastic masked reconstruction and artifact denoising. While these aim to simulate common structural incompleteness and sensor noise, their sufficiency in covering all possible or more extreme types of missing data and distortion patterns could be further explored.

4.	Although Adaptive λ-Calibration addresses the balancing of auxiliary task weights, the MAML framework can be sensitive to hyperparameter choices. Further discussion on the selection strategy for these hyperparameters and their impact on final performance would be beneficial.

---

> ### Author Rebuttal · Authors · 2025-07-30
>
> We thank Reviewer PV7J for the recognition and constructive comments.
>
> # Q1: Bi-Aux Units Visualization
>
> We sincerely thank the reviewer for this suggestion. We agree that additional visualizations—such as attention maps, feature activations, and region-wise responses—will clarify how Bi-Aux Units focus on specific input regions and improve interpretability. We will include these in the revision and welcome further feedback.
>
> # Q2: Computational Cost
> In Rebuttal Table A, we report computational cost in terms of model parameters, FLOPs, and ms/sample. As shown in the table, our model is lightweight compared to existing ones, while still achieving the best performance.
> TTA adds a relatively small amount of overhead to the inference time since most gains are obtained within the first few adaptation steps. The inference time is reported under three adaptation steps.
> Please note that ms/sample is reported using one Nvidia V100.
>
> **Rebuttal Table A**. Computational cost.
>
> | Method       | Params (M) | FLOPs (G) | Inference (ms/sample) |
> |--------------|------------|-----------|-----------------------|
> | GRNet        | 76.71      | 40.44     | 45                    |
> | SnowflakeNet | 19.32      | 10.32     | 19                    |
> | PoinTr       | 30.9       | 10.41     | 22                    |
> | ProxyFormer  | 12.16      | 9.88      | 42                    |
> | CRA-PCN      | 7.46       | 8.13      | 27                    |
> | Ours (w/o TTA)        | 10.92      | 9.26      | 21 (w/o TTA)          |
> | Ours (w/  TTA)        | 10.92      | 9.26      | 36 (w/ TTA)           |
>
>
> # Q3: Evaluation under Extreme Patterns
>
> We thank the reviewer for this important question. We explicitly evaluate the robustness of our approach on both synthetic and real-world datasets under severe structural incompleteness and sensor degradation. **Visualizations of extreme patterns are provided in Supplementary Sec. C (Line 911).**
> * On ShapeNet **(synthetic)**, we tested with up to 75\% of input points missing across 55 categories, as reported in Tables 2–3 of the main paper (CD-H metric).
> * On KITTI **(real-world)**, the input is even sparser (averaging 440 points, or 5.4\% of the completion target, with some samples below 10 points), where our model still achieves strong performance.
>
> Importantly, TTA with meta-auxiliary learning provides a **general framework**: the auxiliary task is not restricted to enumerating all possible failure modes. Instead, it can be **tailored to different objectives as needed**, fundamentally overcoming the limitations of static, pre-defined strategies. This enables dynamic, per-sample adaptation to previously unseen patterns, allowing our model to robustly recover fine details where traditional static training or enumerative augmentation would fail.
>
> # Q4: Hyperparameter Choices
>
> We appreciate the reviewer’s concern regarding hyperparameter sensitivity. Please see **Supplementary Sec. A4 (Lines 867–881)**, where we found that the number of **gradient steps** is the most impactful factor and have provided a thorough analysis. Furthermore, our adaptive $\lambda$-calibration with MAML is designed to **alleviate the need for expensive manual hyperparameter tuning** by learning suitable hyperparameters during meta-training. As such, the learning rate simply follows the initial task-specific setting.
>
>
> **[Dear Reviewer PV7J]** Thank you again for your rigorous and constructive review, which has helped us further improve the quality of our work.

---

> > ### Comment · Reviewer_PV7J · 2025-08-04
> >
> > The authors addressed my concerns and questions.  I will maintain my rating

---

> ### Author Response · Authors · 2025-08-07
>
> **Dear Reviewer PV7J**,
> thank you very much for your positive response. We appreciate your feedback and are grateful that your concerns and questions have been addressed. We are always ready to discuss any further concerns you may have.

---

### Official Review · Reviewer_SjCL · 2025-07-01

**Clarity:** 3
**Significance:** 3
**Originality:** 3
**Rating:** 4
**Confidence:** 5

**Summary:**

To address the domain gap between training data and application, this paper proposed a time-time adaptation method for point cloud completion, named PointMAC. This method optimizes the completion model under two self-supervised auxiliary objectives that simulate structural and sensor-level incompleteness. This is also the first work to apply meta-auxiliary test-time adaptation to point cloud completion.

**Questions:**

1. The Limitations and Future Work part mainly discussed the future work. What is the limitation about this work in the object-level completion setting. Or in other words, under what kind of setting this work may fail?
2. Please provide some stats for computational cost.

**Ethical Concerns:**

["NO or VERY MINOR ethics concerns only"]

**Final Justification:**

My final justification is maintaining my positive rating.

**Limitations:**

yes

**Quality:**

3

**Strengths And Weaknesses:**

Strengths:
1. The first method to apply meta-auxiliary test-time adaptation to point cloud completion.
2. good performance

Weaknesses:
1. cherry-pick concern: Figure 1 is not a convincing proof or illustration of the claim "model tends to generate what we refer to as generic completions". The plane tail structure from your method can be a cherry-picked result. I am not sure why you do adaptation with supervision constructed from a plane without tail and can give the model information about tails. Therefore, it may just be a cherry-picked result.

2. be more specific about the claim: On the other hand, it is validated by many sources that deterministic method tend to produce average prediction (generic completions in the point cloud completion case) while probabilistic method can produce results with sharp and clean features. See Figure 2. of Diffusion Models as Masked Autoencoders. Therefore, I think the authors need to be more specific about this claim.

3. concern about application and significance: it would be also important for a completion method to be fast and real-time. Therefore, the frame per second would be an important metric. The authors should provide some about the computation cost to obtain the good performance. On the one hand, lost real-time processing capability means it can only be applied in limited uses cases, on the other hand, the performance is kind of incremental (small improvement in Table 2 and 3), people need to consider whether it worth doing the proposed TTA.

---

> ### Author Rebuttal · Authors · 2025-07-30
>
> We sincerely thank Reviewer SjCL for constructive comments.
> > **Question**
> # Q1: Limitations and Failure Cases
>
> While TTA improves completion in many challenging cases as compared with existing approaches, it still has inherent limitations under certain conditions. Specifically, our sample-specific TTA is unable to **“hallucinate”** plausible fine-grained geometry beyond what is **already encoded in the model’s priors**. Therefore, when objects have extremely complicated geometric structures and the input point cloud provides minimal information, the adaptation process may still overfit to the limited input, resulting in blurry completions, especially for categories like birdhouses.
>
> For future work, we intend to explore TTA for **post-training quantized (PTQ) generative models** [1], aiming to enhance completion accuracy in challenging scenarios while maintaining the diversity of generated results. We will clarify this limitation and discuss these potential directions in the revised manuscript.
> # Q2: Computational Cost
> In Rebuttal Table A, we report computational cost in terms of model parameters, FLOPs, and ms/sample. As shown in the table, our model is lightweight compared to existing ones, while still achieving the best performance.
> TTA adds a relatively small amount of overhead to the inference time since most gains are obtained within the first few adaptation steps. The inference time is reported under three adaptation steps.
> Please note that ms/sample is reported using one Nvidia V100.
>
> **Rebuttal Table A**. Computational cost.
>
> | Method       | Params (M) | FLOPs (G) | Inference (ms/sample) |
> |--------------|------------|-----------|-----------------------|
> | GRNet        | 76.71      | 40.44     | 45                    |
> | SnowflakeNet | 19.32      | 10.32     | 19                    |
> | PoinTr       | 30.9       | 10.41     | 22                    |
> | ProxyFormer  | 12.16      | 9.88      | 42                    |
> | CRA-PCN      | 7.46       | 8.13      | 27                    |
> | Ours (w/o TTA)        | 10.92      | 9.26      | 21 (w/o TTA)          |
> | Ours (w/  TTA)        | 10.92      | 9.26      | 36 (w/ TTA)           |
>
> > **Weakness**
> # W1: Cherry-picking Concern on Figure 1
> Figure 1 shows a typical example from the PCN test set. The improvement can also be observed in the results shown in Fig. 4 and Fig. 5 of the main paper.
> They illustrate the primary benefit of our sample-specific adaptation. We believe the underlying reason for getting the improvement is that the inner-loop optimization in TTA with MAML updates model parameters along the gradient of the self-supervised loss, gradually steering the model output toward globally and semantically plausible completions (e.g., an airplane with a reasonable tail) for each test sample. The outer-loop meta-optimization, driven by the completion task, ensures that the auxiliary branches provide effective adaptation signals. Please refer to **Alg. 2 in Supplementary A.3.2 (Lines 846–866)**.
>
> This represents a **paradigm shift from traditional static**, **one-size-fits-all inference**. As a result, improvements are **consistent rather than cherry-picked**. This is also validated in Tables 1–5 **(across all benchmark datasets)** of the original submission. We will include more compelling examples in the final version to further demonstrate its generalization ability.
> # W2: Specifics of the Claim (Conventional Generic Completion vs. Our Sample-specific Completion with TTA)
> We appreciate the reviewer’s insightful suggestion and agree that it is important to clarify this distinction. Basically, in point cloud completion, there are two main types of models: deterministic models and probabilistic models (e.g., diffusion models). In this work, we study deterministic models that usually yield averaged (generic) predictions in existing works due to mean regression. That motivates us to develop a sample-specific adaptation strategy to predict the geometry details for each test sample.
> # W3: Concern about Application and Significance
>
> As shown in Rebuttal Table A, our method consistently achieves improved accuracy while maintaining a reasonable computational cost. Moreover, for **real-world deployment**, where test and training data often differ significantly due to changes in sensors, objects, and other factors, **domain shift** can lead to performance degradation when deploying a fixed, pre-trained model. On the contrary, our **TTA approach enables continuous personalization to new environments and samples** by dynamically adapting to each test instance. This claim is also **validated in our cross-dataset evaluation** as shown in Table 5 of the main paper and Fig. 7 in the Supplementary (Line 911)
>
> [1] Post-training quantization on diffusion models.
>
>
> **[Dear Reviewer SjCL]** We value your feedback and look forward to discussing any further concerns you may have.

---

> > ### Comment · Reviewer_SjCL · 2025-08-04
> >
> > I appreciate the authors' efforts in providing additional results and explanations. Most of my concerns have been addressed, and I will maintain my positive rating.

---

> ### Author Response · Authors · 2025-08-07
>
> **Dear Reviewer SjCL**,
> thank you very much for your constructive comments and positive rating. We appreciate your acknowledgment that most of your concerns have been addressed. If you have any further questions or suggestions, we would be grateful for the opportunity to discuss them.

---

### Official Review · Reviewer_sY8X · 2025-07-03

**Clarity:** 2
**Significance:** 3
**Originality:** 3
**Rating:** 4
**Confidence:** 4

**Summary:**

PointMAC proposes a novel test-time adaptation framework for point cloud completion that enables per-sample refinement without requiring labels. Unlike existing models that rely on static inference and training-set priors, PointMAC adapts its shared encoder at test time using two self-supervised auxiliary tasks—Stochastic Masked Reconstruction and Artifact Denoising—which simulate occlusions and sensor noise, respectively. A meta-learning strategy based on MAML ensures that these auxiliary adaptations consistently benefit the main completion task, while an Adaptive λ-Calibration mechanism dynamically balances the influence of auxiliary losses. Extensive experiments across synthetic, simulated, and real-world datasets show that PointMAC achieves state-of-the-art performance by producing sample-specific completions that generalize well beyond training data.

**Questions:**

* What is the computational cost (e.g., runtime per sample, FLOPs, or memory overhead) of the proposed test-time adaptation step?

* Have you attempted to apply your test-time adaptation module to any existing static inference baselines (e.g., PCN or PoinTr)? If not, could you comment on its feasibility and any practical challenges?

* Consider revising Figures 2 and 3 to better distinguish between shared components, auxiliary branches, and adaptation steps. A clearer flow would help readers follow the interactions between modules more easily.

* Please consider expanding the related work section to include recent test-time adaptation methods specifically designed for 3D point clouds.

**Ethical Concerns:**

["NO or VERY MINOR ethics concerns only"]

**Final Justification:**

The rebuttal satisfactorily addresses my main concerns. The authors provided computational cost analysis (Table A), demonstrated plug-and-play applicability on PoinTr and SnowflakeNet (Table B), and committed to improving figures, expanding the related work to contain recent 3D point cloud TTA methods (e.g., 3DD-TTA, CloudFixer), and releasing code. These clarifications strengthen the paper’s technical soundness. I maintain my borderline accept recommendation, now with higher confidence.

**Limitations:**

Yes.

**Paper Formatting Concerns:**

No.

**Quality:**

3

**Strengths And Weaknesses:**

__Strengths:__
* The paper is technically solid and methodologically well-designed.
* The use of self-supervised auxiliary tasks (masked reconstruction and denoising) is thoughtfully designed to simulate real-world data challenges
* The experimental evaluation is extensive, covering synthetic, simulated, and real-world datasets.
* The combination of sample-specific refinement and MAML in the context of 3D point cloud completion is impactful.

__Weaknesses:__
* The block diagrams could be made more modular and annotated to better illustrate the interactions between the encoder, auxiliary branches, and meta-learning loops.
* The pipeline involves several components (meta-learning, auxiliary decoders, adaptive balancing), making reproducibility difficult without access to code. More implementation details or pseudocode would help.
* The paper lacks analysis of the computational overhead introduced by test-time adaptation. Reporting runtime or FLOPs would clarify its practicality in real-time settings.
* The proposed TTA method could likely be applied to static baselines (e.g., PCN, PoinTr) with some integration. Demonstrating this would show broader utility.
* The paper could be strengthened by referencing recent point cloud-specific TTA methods like:
  - “Test-Time Adaptation of 3D Point Clouds via Denoising Diffusion Models”
  - “CloudFixer: Test-Time Adaptation for 3D Point Clouds via Diffusion-Guided Geometric Transformation”

---

> ### Author Rebuttal · Authors · 2025-07-30
>
> We sincerely thank Reviewer sY8X for the valuable suggestions.
> # Q1: Computational Cost
> In Rebuttal Table A, we report computational cost in terms of model parameters, FLOPs, and ms/sample. As shown in the table, our model is lightweight compared to existing ones, while still achieving the best performance.
> TTA adds a relatively small amount of overhead to the inference time since most gains are obtained within the first few adaptation steps. The inference time is reported under three adaptation steps.
> Please note that ms/sample is reported using one Nvidia V100.
>
> **Rebuttal Table A**. Computational cost.
>
> | Method       | Params (M) | FLOPs (G) | Inference (ms/sample) |
> |--------------|------------|-----------|-----------------------|
> | GRNet        | 76.71      | 40.44     | 45                    |
> | SnowflakeNet | 19.32      | 10.32     | 19                    |
> | PoinTr       | 30.9       | 10.41     | 22                    |
> | ProxyFormer  | 12.16      | 9.88      | 42                    |
> | CRA-PCN      | 7.46       | 8.13      | 27                    |
> | Ours (w/o TTA)        | 10.92      | 9.26      | 21 (w/o TTA)          |
> | Ours (w/  TTA)        | 10.92      | 9.26      | 36 (w/ TTA)           |
>
> # Q2: Evaluating TTA on General Backbones
> We agree that such experiments are essential for demonstrating the broader applicability of our method. As shown in Rebuttal Table B, we have **validated the plug-and-play effectiveness of our TTA** on PoinTr and SnowflakeNet, to demonstrate practical generalization.
>
> **Rebuttal Table B**. TTA on general backbones.
> | Backbone (CD-ℓ₁)   | Method             | Plane | Cabinet | Car  | Chair | Lamp | Couch | Table | Boat | Avg  |
> |---------------------|--------------------|-------|---------|------|-------|------|-------|-------|------|------|
> | PoinTr              | baseline           | 4.75  | 10.47   | 8.68 | 9.39  | 7.75 | 10.93 | 7.78  | 7.29 | 8.38 |
> | PoinTr              | w/ Bi-Aux          | 4.61  | 10.15   | 8.39 | 9.11  | 7.57 | 10.63 | 7.53  | 7.09 | 8.14 |
> | PoinTr              | w/ (Bi-Aux + TTA)  | 4.59  | 9.98    | 8.26 | 8.95  | 7.41 | 10.37 | 7.39  | 6.95 | 7.98 |
> | SnowflakeNet        | baseline           | 4.29  | 9.16    | 8.08 | 7.89  | 6.07 | 9.23  | 6.55  | 6.40 | 7.21 |
> | SnowflakeNet        | w/ Bi-Aux          | 4.13  | 8.94    | 7.80 | 7.65  | 5.93 | 8.99  | 6.34  | 6.22 | 7.00 |
> | SnowflakeNet        | w/ (Bi-Aux + TTA)  | 4.08  | 8.70    | 7.69 | 7.57  | 5.86 | 8.75  | 6.20  | 6.09 | 6.86 |
>
>
> # Q3: Figure Clarity and Workflow
> We will revise Figures 2 and 3 accordingly in the final version. Please also refer to **Algorithm 2 in Supplementary A.3.2** for the detailed Meta-Auxiliary Training process.
> We will release the source code upon acceptance.
>
> # Q4: Expand Related Work
>
>
> We agree that adding recent TTA methods for 3D point clouds, such as 3DD-TTA [1] and CloudFixer [2], will strengthen our related work section.
>
> We will include these approaches, which primarily focus on **input-level adaptation** (e.g., modifying the test data distribution via diffusion or denoising). In contrast, **our model adopts a model-level adaptation** strategy, dynamically updating model parameters for each test sample. We will clarify these distinctions and expand our discussion in the revised manuscript. We welcome any further recommendations from the reviewer.
>
>
> [1] Test-time adaptation of 3d point clouds via denoising diffusion models.
>
> [2] Cloudfixer: Test-time adaptation for 3d point clouds via diffusion-guided geometric transformation.
>
> **[Dear Reviewer sY8X]** We value your feedback and look forward to discussing any further concerns you may have.

---

> ### Author Response · Authors · 2025-08-07
>
> **Dear Reviewer sY8X**,
> we hope that our experiments and analyses have addressed your concerns. Thank you very much for your constructive suggestions. If you have any additional concerns, we would sincerely appreciate the opportunity to discuss them.

---

> > ### Comment · Reviewer_sY8X · 2025-08-08
> >
> > The rebuttal satisfactorily addresses my main concerns. The authors provided computational cost analysis (Table A), demonstrated plug-and-play applicability on PoinTr and SnowflakeNet (Table B), and committed to improving figures, expanding the related work to contain recent 3D point cloud TTA methods (e.g., 3DD-TTA, CloudFixer), and releasing code. These clarifications strengthen the paper’s technical soundness. I maintain my borderline accept recommendation, now with higher confidence.

---

> > > ### Author Response · Authors · 2025-08-08
> > >
> > > **Dear Reviewer sY8X**, we are very pleased to have addressed your concern. Thank you for your positive feedback.

---

### Note · Authors · 2025-08-12

Dear PCs, SACs, ACs, and reviewers,

We sincerely thank the reviewers for their thoughtful feedback and constructive discussions. We are pleased that **all concerns were addressed**, resulting in **positive ratings from all reviewers**. We greatly appreciate both the recognition of our work’s significance and the valuable suggestions, which we will incorporate into future revisions.

This reflects a shared appreciation for our proposed test-time adaptation (TTA) framework with meta-auxiliary learning, which establishes a **general paradigm** for point cloud completion. Unlike traditional methods that use static inference or focus on encoder/decoder design, our method adapts at test time for sample-specific completion and achieves SOTA performance across all benchmarks.

We would be grateful if AC would consider the potential of this paradigm to contribute both practically and conceptually, and to inspire further research in point cloud understanding.

---

### Decision · Program_Chairs · 2025-09-17

**Decision:**

Accept (poster)

**Comment:**

This paper proposes PointMAC, a novel meta-learning framework for robust test-time adaptation in point cloud completion. The motivation is to address the limitations of existing models that 1) perform static inference and 2) require training-learned inductive biases. Extensive experiments on several benchmarks show its effectiveness.

All reviewers agreed that the paper proposes a solid approach and the evaluations are satisfactory. The AC agrees with the recommendation to accept this paper.